# SIMPC: Learning Self-Induced Mirror-Point Consistency for Unsupervised Point Cloud Denoising

**Chengwei Zhang** [1]  **Xueyi Zhang** [2]  **Tao Jiang** [1]  **Xinhao Xu** [1]  **Wenjie Li** [1]  **Fubo Zhang** [1]  **Longyong Chen** [1]

## Abstract

In point clouds, noise directly perturbs point coordinates that encode both spatial location and geometry, making one-to-one correspondence construction more challenging than in images. Existing methods impose statistical mappings across noisy variants via noise or optimal transport, but suffer from correspondence ambiguity. In this work, we propose **S**elf-**I**nduced **M**irror-**P**oint **C**onsistency **(SIMPC)** to learn deterministic correspondences between points and the underlying surface in an unsupervised manner. For each noisy point, SIMPC generates a mirror-point on the opposite side of the underlying surface, guided by geometric priors during the denoising process. By encouraging consistency between the denoising targets of the original point and its mirror counterpart, SIMPC effectively localizes the position of underlying surface. Extensive experiments on synthetic and real-world datasets demonstrate that SIMPC significantly outperforms state-of-the-art unsupervised methods and surpasses several strong supervised counterparts.

## 1. Introduction

Point clouds are a fundamental 3D representation but are often croupted by noise in real-world acquisition, which degrades the performance of downstream tasks such as surface reconstruction and semantic understanding (Huang et al., 2024; Ren et al., 2022). Consequently, point cloud denoising is widely adopted as a preprocessing step to recover geometric structures and improve the robustness of representation extraction (Li et al., 2024; Sun et al., 2023).

[1]National Key Laboratory of Microwave Imaging, Aerospace Information Research Institute, Chinese Academy of Sciences, Beijing, China [2]School of Computing, National University of Singapore, Singapore. Correspondence to: Longyong Chen <chenly@aircas.ac.cn>.

*Proceedings of the $43^{rd}$ International Conference on Machine Learning*, Seoul, South Korea. PMLR 306, 2026. Copyright 2026 by the author(s).

Despite notable progress, supervised point cloud denoising methods heavily depend on paired noisy-clean data, which are commonly synthesized from CAD models. Building such datasets requires substantial manual effort to increase training data diversity, thereby ensuring generalization across different geometries and scenes (Chen et al., 2022a). By contrast, modern LiDAR systems can continuously acquire large-scale noisy point clouds from diverse environments. This practical advantage naturally motivates unsupervised point cloud denoising, which seeks to learn denoising without relying on paired supervision.

For 2D images, as in Fig 1(a), multiple corrupted observations admit pixel-wise correspondences, such that noisy values at the same pixel location across different observations can be regarded as random realizations drawn from a distribution centered at a clean pixel value. In contrast, point clouds do not admit fixed spatial indexing, and noise perturbs point coordinates that encode both location and geometry, breaking correspondence consistency across noisy observations. Hence, the pixel-wise assumptions of unsupervised image denoising do not apply to point clouds.

Recent unsupervised point cloud denoising methods leverage statistical mappings across noisy variants and can be broadly categorized into: **(1) Noise-based methods** (e.g., Noise4Denoise (Wang et al., 2024b), Noise2Score3D (Wei et al., 2025)), as in Fig 1(b), generate noisier variants by injecting random noise and learn an inverse noise mapping to regress toward a cleaner state. However, the induced correspondences are purely driven by stochastic noise and are not explicitly pointing to the target surface. **(2) EMD-based methods** (e.g., NoiseMap (Ma et al., 2023), U-CAN (Zhou et al., 2025a)), as in Fig 1(c), employ Earth Mover's Distance to impose optimal transport between denoised outputs and noisy variants, yielding more structured correspondences than noise-based methods. Nevertheless, such transport operates at the point-set distribution level and does not explicitly encourage point-level consistency with target surface. *Consequently, the resulting mappings suffer from correspondence ambiguity, where points with one-to-one correspondence across noisy variants are not guaranteed to represent different observations of the same target surface.*

To address this challenge, we propose Self-Induced Mirror-

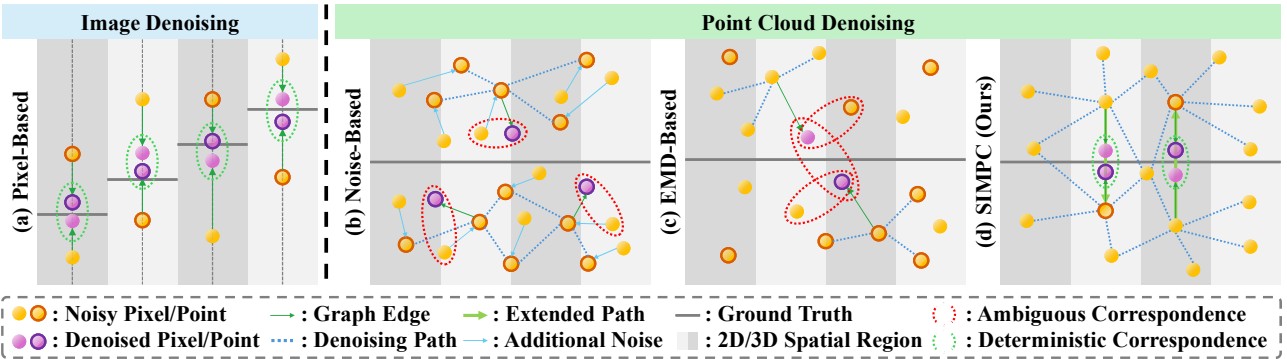

*Figure 1.* An illustration of the differences between image and point cloud denoising. Points w/ and w/o darker outlines indicate different noisy variants. For images, pixel-based indexing in (a) enables establishing correspondences between noisy variants associated with the same ground truth. Existing point cloud denoising methods, such as (b) Noise-based and (c) EMD-based approaches, establish ambiguous correspondences. By contrast, our proposed (d) SIMPC extracts geometric priors during the denoising process to generate mirror-points with deterministic correspondences, and further localizes the position of the underlying surface by learning consistent denoising targets.

Point Consistency (SIMPC), a novel unsupervised framework that learns deterministic one-to-one correspondences. The key insight of SIMPC is to establish correspondences that are self-induced from a single noisy point, as in Fig 1(d), rather than matching from independently sampled noisy observations. Specifically, SIMPC introduces a Mirror-Point Generation Module (MPGM) to extract geometric priors during denoising, which encode a coarse localization of the target surface. Based on these priors, a mirror-point is generated on the opposite side of the target surface, forming a deterministic correspondence between points associated with the same target surface, while exhibiting different neighboring noise states. We further introduce a Mirror-Point Consistency Loss (MPCL) to encourage consistency between the denoising targets of the original point and its mirror counterpart, enabling SIMPC to produce more accurate and stable denoising outputs. Extensive experiments on synthetic datasets with both Gaussian and non-Gaussian noise, as well as on more challenging real-world noisy datasets, demonstrate that SIMPC consistently outperforms state-of-the-art unsupervised point cloud denoising methods and even surpasses several strong supervised counterparts.

Our main contributions are summarized as follows:

- We propose SIMPC, a novel unsupervised point cloud denoising framework that establishes deterministic one-to-one correspondences between noisy points associated with the same target surface.

- We design MPGM to extract geometric priors during denoising for generating self-induced mirror counterparts of original points, and introduce MPCL to encourage consistent denoising targets between the point pairs.

- Extensive experiments demonstrate that SIMPC achieves state-of-the-art performance among unsupervised methods on both synthetic and real-world datasets, and even surpasses several strong supervised approaches.

## 2. Related Work

### 2.1. Supervised Point Cloud Denoising

Supervised methods rely on clean point clouds to learn denoising, thereby achieving high-quality denoising results. In practice, existing approaches either denoise the entire point cloud by treating points independently (de Silva Edirimuni et al., 2023a; Rakotosaona et al., 2020; Wang et al., 2024a; Wei et al., 2024; Zhang et al., 2020), or split the input into local patches (de Silva Edirimuni et al., 2023b; 2024; Guo et al., 2025; Luo & Hu, 2021) for parallel processing. Within these frameworks, denoising can be realized through point resampling (Li & Sheng, 2023; Luo & Hu, 2020; Wang et al., 2023), displacement prediction (de Silva Edirimuni et al., 2023b; Luo & Hu, 2021), or direct coordinate regression (Mao et al., 2022; 2024). Correspondingly, supervision signals are exploited in various forms, including constraining denoising vectors (Chen et al., 2024; de Silva Edirimuni et al., 2023b; Liu et al., 2025; Vogel et al., 2025; Zhou et al., 2025b), or directly minimizing similarity metrics between denoised and clean point clouds, e.g., the Chamfer Distance (Chen et al., 2022b; Guo et al., 2025; Zhang et al., 2024) and the Earth Mover's Distance (Luo & Hu, 2020; Mao et al., 2022; 2024). Moreover, iterative denoising is a prevalent paradigm for progressively removing residual noise, leading to more convergent results. Specifically, some approaches stack multiple denoising blocks and feed the output into subsequent iterations (Chen et al., 2024; de Silva Edirimuni et al., 2023b). Building upon this, alternative approaches further model the denoising process in a latent space to achieve more stable denoising (Wei et al., 2024; Zhang et al., 2024). Despite these advances, supervised methods exhibit significant performance degradation under limited training data (Chen et al., 2022a).

## 2.2. Unsupervised Point Cloud Denoising

Unsupervised point cloud denoising is primarily inspired by advances in unsupervised image denoising, which eliminate the need for clean supervision by exploiting statistical properties of noise (Batson & Royer, 2019; Huang et al., 2021; Krull et al., 2019; Lehtinen et al., 2018). In an early stage, several attempts leverage density-based priors to learn the position of the target surface (Hermosilla et al., 2019; Luo & Hu, 2020; 2021). However, the stability of density estimation is fundamentally hindered by the sparsity and discreteness of point clouds. As a result, recent methods increasingly focus on learning statistical mappings across noisy variants. Among them, noise-based approaches, such as Noise4Denoise (Wang et al., 2024b) and Noise2Score3D (Wei et al., 2025), generate noisier variants by injecting random noise and learn inverse denoising mappings toward cleaner states. During inference, these methods extend the predicted denoising vectors to approach the target surface by either fixed (Moran et al., 2020) or target-adaptive (Kim & Ye, 2021) scaling. Nevertheless, such noise-driven extrapolation relies on assumptions about noise intensity or distribution, which are often unavailable in real-world point clouds, leading to unstable denoising behavior and degraded performance (Huang et al., 2021). Beyond noise-based strategies, another line of research explores optimal transport to align noisy observations with denoised outputs. Specifically, NoiseMap (Ma et al., 2023; Zhou et al., 2024) fits per-instance signed distance functions using EMD-based local correspondences, but suffers from severe overfitting, high computational cost, and limited generalization due to instance-wise optimization. Furthermore, U-CAN (Zhou et al., 2025a) encourages consistency across denoised outputs from different noisy inputs, yet still struggles to accurately localize the underlying surface.

## 3. One-to-One Point Correspondence

Given a 3D shape or scene $\mathcal{S}$, we consider multiple corrupted observations $\mathcal{X} = \{\mathcal{X}^m \in \mathbb{R}^{N \times 3} \mid m \in [1, M]\}$, sampled from $\mathcal{S}$ under a noise level $\sigma$, where $M$ denotes the number of observations and $N$ is the number of points. In this setting, existing methods typically establish one-to-one point correspondences across different observations to enable unsupervised point cloud denoising. According to the mechanism used to construct such correspondences, these methods can be broadly categorized into two types: *Noise-based* and *EMD-based* methods.

### 3.1. Noise-based methods

Noise-based methods construct a noisier point cloud by injecting an additional random noise vector $\mathbf{u} \sim \mathcal{N}(\mathbf{0}, \Delta\sigma)$ into a noisy point cloud $\mathcal{X}_a \in \mathcal{X}$, yielding $\hat{\mathcal{X}}_a = \mathcal{X}_a + \mathbf{u}$, whose noise level is increased accordingly. Subsequently,

the denoiser parameters $\theta$ are optimized to predict the inverse noise vector $-\mathbf{u}$ as:

$$\min_{\theta} \ \text{MSE}\left(D(\hat{\mathcal{X}}_a), -\mathbf{u}\right), \qquad (1)$$

where $D(\cdot)$ produces a point-wise denoising vector, and $\text{MSE}(\cdot, \cdot)$ denotes the mean squared error. Through the one-to-one correspondence established by the inverse noise vector, the model learns a statistical mapping from a higher-noise state to a lower-noise state.

### 3.2. EMD-based methods

In contrast, EMD-based methods introduce denoised point clouds as intermediate states and establish point-wise correspondences across different noisy observations by minimizing the Earth Mover's Distance (EMD). Specifically, given two noisy point clouds $\mathcal{X}_a, \mathcal{X}_b \in \mathcal{X}$, the optimization objective is formulated as:

$$\min_{\theta} \Bigg( \underbrace{\text{EMD}(\mathcal{X}_a + D(\mathcal{X}_a), \ \mathcal{X}_b) + \text{EMD}(\mathcal{X}_b + D(\mathcal{X}_b), \ \mathcal{X}_a)}_{\text{Noise-to-Noise Matching}} +$$
$$\underbrace{\text{EMD}(\mathcal{X}_a + D(\mathcal{X}_a), \ \mathcal{X}_b + D(\mathcal{X}_b))}_{\text{Denoising Consistency}} \Bigg),$$

$$(2)$$

where $\text{EMD}(\cdot, \cdot)$ computes the optimal transport cost between two point clouds. The Noise-to-Noise Matching term constrains the distribution of denoised point clouds and mitigates point collapse (Ma et al., 2023), while the Denoising Consistency term encourages consistent denoising results across different noisy observations (Zhou et al., 2025a).

## 4. Self-Induced Mirror-Point Consistency

As illustrated in Fig. 2, our method follows the iterative denoising paradigm adopted in prior works (de Silva Edirimuni et al., 2023b; Wei et al., 2024; Zhang et al., 2024; Zhou et al., 2025a). Following prior work (Zhou et al., 2025a), we select two noisy variants as $\mathcal{X}_a, \mathcal{X}_b \in \mathcal{X}$. We take $\mathcal{X}^0 \in \mathbb{R}^{N \times 3}$ as the initial input to the network, where $\mathcal{X}^0 \in \{\mathcal{X}_a, \mathcal{X}_b\}$. Each noisy variant is processed independently.

### 4.1. Encoder

We first employ an encoder to extract initial point-wise features from the input noisy point cloud $\mathcal{X}^0 = \{x_i^0\}_{i=1}^N$. As illustrated on the left side of Fig. 2, the encoder is implemented as a $T$-layer DGCNN (de Silva Edirimuni et al., 2024; Wang et al., 2019) with $T = 3$, equipped with skip connections across graph convolution layers. At the $t$-th graph convolution layer, we denote the point-wise feature set as $G^t = \{g_i^t\}_{i=1}^N$. We initialize the features with point coordinates, i.e., $G^0 = \mathcal{X}^0$, and progressively lift the feature dimension through the encoder. Given the point-wise

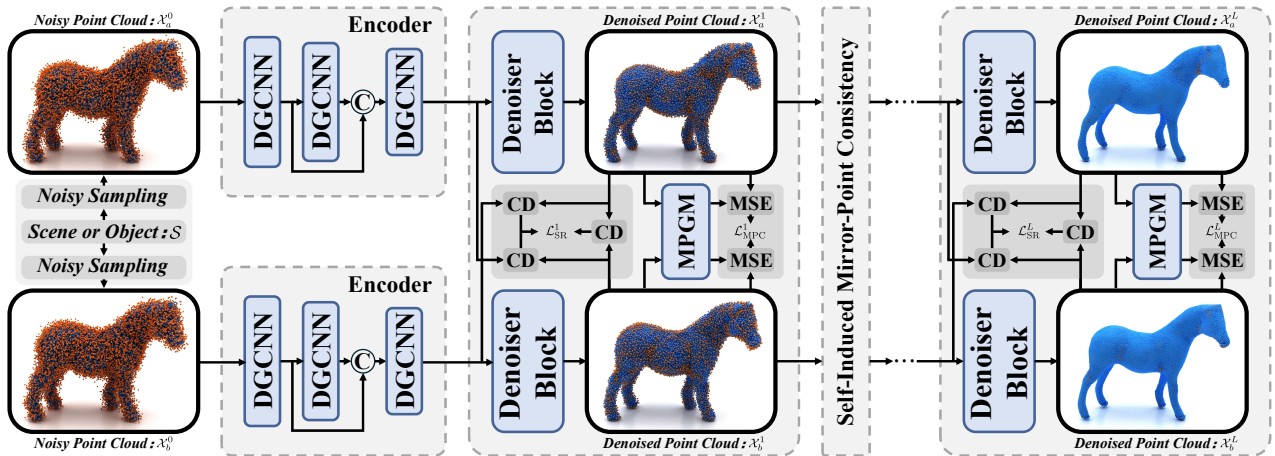

*Figure 2.* An illustration of the proposed **Self-Induced Mirror-Point Consistency (SIMPC)** framework. Given two noisy point clouds, we perform iterative denoising using shared denoiser blocks. Unlike previous approaches that establish relationships across noisy variants through noise injection or EMD-based alignment, we employ the Chamfer Distance to provide a basic similarity constraint between variants. Furthermore, within each variant, we introduce a **Mirror-Point Generation Module (MPGM)** and **Mirror-Point Consistency Loss (MPCL)** to learn deterministic one-to-one correspondences.

feature $g_i^t$, we construct a dynamic neighborhood in the feature space as $\mathcal{N}_i^t = \mathrm{KNN}(g_i^t, G^t, k)$ with $k = 32$, which contains the $k$ nearest neighbors of point $i$. We then update the point-wise feature over the dynamic graph:

$$g_i^{t+1} = h(g_i^t) + \sum_{j \in \mathcal{N}_i^t} h\big([g_i^t \,\|\, (g_j^t - g_i^t)]\big), \qquad (3)$$

where $[\cdot \,\|\, \cdot]$ denotes feature concatenation and $h(\cdot)$ is parameterized by an MLP. After $T$ layers, we obtain the encoded point-wise features $U^0 = \{u_i^0\}_{i=1}^N \in \mathbb{R}^{N \times C}$, where we set $U^0 = G^T$, $u_i^0$ is the initial feature of point $i$, and $C = 256$ is the feature dimension.

### 4.2. Denoiser Block

**Point Self Attention.** We first apply a Point Self-Attention (PSA) layer (Pan et al., 2021) to refine point-wise features and enhance local geometric perception. At the $l$-th denoiser block, given the point-wise feature $u_i \in U^l$ and its corresponding coordinate $x_i \in \mathcal{X}^l$, we construct a neighborhood index set as $\hat{\mathcal{N}}_i = \mathrm{KNN}(x_i, \mathcal{X}^l, k)$, which contains the $k$ nearest neighbors of point $i$ and their associated features $u_j$, $j \in \hat{\mathcal{N}}_i$. The attention weights are computed as:

$$\alpha_{ij} = h\Big([h(u_i), \{h(u_j)\}_{j \in \hat{\mathcal{N}}_i}]\Big), \qquad (4)$$

and the aggregated feature is obtained by:

$$f_i = \sum_{j \in \hat{\mathcal{N}}_i} \alpha_{ij} \odot h(u_j), \qquad (5)$$

where $F^l = \{f_i\}_{i=1}^N \in \mathbb{R}^{N \times C}$ denotes the refined feature at the $l$-th block, and $\odot$ indicates element-wise multiplication.

The above process can be compactly formalized as:

$$f_i = \mathrm{PSA}\big(u_i, \{u_j\}_{j \in \hat{\mathcal{N}}_i}\big). \qquad (6)$$

The output feature $F^l$ of the PSA layer in the current denoiser block is then fed as the input feature of the next denoiser block, i.e., $U^{l+1} = F^l$.

**Decoder.** At the $l$-th denoiser block, the decoder takes the point-wise feature $f_i$ as input. It consists of a sequence of fully connected layers interleaved with ReLU activations. The decoder outputs a point-wise denoising vector as:

$$d_i = \mathrm{Dec}(f_i). \qquad (7)$$

where $d_i \in \mathbb{R}^3$ denotes the denoising vector, which is normalized by a $\mathtt{tanh}$ activation and used to update the point coordinate at the current block.

**Denoising Process.** At the $l$-th denoiser block, the point coordinates are further refined based on the output of the previous block $\mathcal{X}^{l-1} = \{x_i^{l-1}\}_{i=1}^N$ as:

$$x_i^l = x_i^{l-1} + d_i^l. \qquad (8)$$

By iteratively applying the above denoising blocks, we obtain a point cloud sequence $\mathcal{D} = \{\mathcal{X}^0, \mathcal{X}^1, \dots, \mathcal{X}^L\}$, where $L$ denotes the number of denoiser blocks, which is set to $L = 2$ in our implementation to balance computational cost and denoising performance. Here, $\mathcal{X}^0$ is the original noisy point cloud, and $\mathcal{X}^l$, $1 \le l \le L$, is the denoised point cloud at the $l$-th stage.

### 4.3. Mirror-Point Generation Module

To establish deterministic one-to-one correspondences, we introduce a Mirror-Point Generation Module (MPGM),

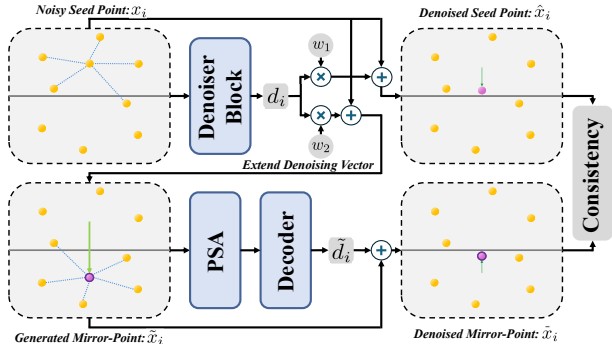

*Figure 3.* An illustration of the Mirror-Point Generation Module. Given a noisy and denoised seed point $x_i$ and $\hat{x}_i$, we extend the denoising vector to generate a mirror-point $\tilde{x}_i$ on the opposite side of the underlying surface. We then apply PSA and the decoder to perform denoising under the new neighborhood and obtain the denoised mirror-point $\bar{x}_i$. Finally, for the two points $\hat{x}_i$ and $\bar{x}_i$ with deterministic correspondence, we encourage consistency between them to facilitate more accurate learning of the underlying surface.

whose architecture is illustrated in Fig. 3. Instead of constructing correspondences across different noisy observations, MPGM operates on each noisy point individually and generates self-induced mirror counterparts guided by the denoising process. At the $l$-th denoiser block, we treat $x_i$ as a noisy seed point together with its corresponding denoising vector $d_i$, and first obtain a normally denoised seed point as $\hat{x}_i = x_i + w_1 d_i$, where $w_1 = 1$ controls the standard denoising strength. To construct a mirror counterpart on the opposite side of the underlying surface, we further extend the denoising vector with a larger scaling factor $w_2 > w_1$ and obtain the generated mirror-point as $\tilde{x}_i = x_i + w_2 d_i$, where we set $w_2 = 2$ to ensure geometric symmetry. In the appendix, we provide a theoretical analysis of the deterministic and geometric symmetry design. This generated mirror-point $\tilde{x}_i$ is expected to lie on the opposite side of the underlying surface and thus serves as the mirror input of $x_i$. Based on the new coordinate $\tilde{x}_i$, we recompute its local neighborhood in the original noisy point cloud (excluding the original noisy point $x_i$) as $\tilde{\mathcal{N}}_i = \text{KNN}(\tilde{x}_i, \mathcal{X}^l \setminus \{x_i\}, k)$. Using the same point-wise feature $u_i$ and the features of its mirror neighbors $\{u_j\}_{j \in \tilde{\mathcal{N}}_i}$, we apply the PSA operator to obtain the refined mirror feature:

$$\tilde{f}_i = \text{PSA}\big([u_i \tilde{x}_i], \{[u_j, x_j]\}_{j \in \tilde{\mathcal{N}}_i}\big). \tag{9}$$

The refined mirror feature $\tilde{f}_i$ is then fed into the decoder to predict the denoising vector of the mirror-point:

$$\tilde{d}_i = \text{Dec}(\tilde{f}_i), \tag{10}$$

and the final denoised mirror-point is obtained as $\bar{x}_i = \tilde{x}_i + \tilde{d}_i$. Through the above procedure, we obtain two denoised points $\hat{x}_i$ and $\bar{x}_i$ that are self-induced from the same noisy observation and are deterministically associated with the same underlying surface location.

## 4.4. Mirror-Point Consistency Loss

We then calculate the Mirror-Point Consistency Loss (MPCL) between the denoised seed point $\hat{x}_i$ and denoised mirror-point $\bar{x}_i$ as:

$$\mathcal{L}_{\text{MPC}} = \sum_{i=1}^{N} \|\hat{x}_i - \bar{x}_i\|_2^2, \tag{11}$$

which encourages the denoising targets of each noisy seed point and its mirror counterpart to converge to a common target surface. In the appendix, we provide a theoretical analysis showing how MPCL promotes consistent denoising targets for accurate surface localization.

## 4.5. Overall Objective

The overall training objective integrates the proposed mirror-point consistency loss with a Chamfer Distance-based similarity regularization. The total loss is defined as:

$$\mathcal{L}_{\text{total}} = \sum_{l=1}^{L} \Big(\mathcal{L}_{\text{MPC}}^l + \mathcal{L}_{\text{SR}}^l\Big), \tag{12}$$

where the mirror-point consistency loss $\mathcal{L}_{\text{MPC}}^l$ construct deterministic one-to-one correspondence between points, while $\mathcal{L}_{\text{SR}}^l$ provides coarse similarity regularization at the point-set level. Specifically, $\mathcal{L}_{\text{SR}}^l$ is formulated as:

$$\mathcal{L}_{\text{SR}}^l = CD(\mathcal{X}_a^l, \mathcal{X}_b^l) + CD(\mathcal{X}_a^{l-1}, \mathcal{X}_b^l) + CD(\mathcal{X}_b^{l-1}, \mathcal{X}_a^l), \tag{13}$$

where $\mathcal{X}^l$ belongs to the corresponding point cloud sequence $\mathcal{D}$, and $CD(\cdot, \cdot)$ denotes the Chamfer distance.

## 5. Experiments

### 5.1. Experimental Setup

**Training Data.** We construct the synthetic training set based on the 40 training shapes from the PUNet (Yu et al., 2018) dataset. Following prior unsupervised denoising protocols (Luo & Hu, 2021; Zhou et al., 2025a), we exclusively use noisy point clouds with Gaussian noise injected at scales ranging from 0.5% to 2% of the bounding sphere radius.

**Testing Data.** We evaluate on 20 test shapes from PUNet (Yu et al., 2018) and 10 shapes from PCNet (Rakotosaona et al., 2020) provided by Score (Luo & Hu, 2021), under 10K and 50K resolutions with Gaussian noise at scales of 1%, 2%, and 3%. We further conduct robustness evaluation on PUNet with four types of non-Gaussian noise following (Chen et al., 2022a). In addition, we report results on real-world datasets, including Paris-Rue-Madame (Serna et al., 2014) and Kinect (Wang et al., 2016).

**Implementation Details.** All experiments are implemented in PyTorch 2.0.0 with CUDA 11.8 and executed on a single

| | | | 10K Points | | | | | | 50K Points | | | | | |
|---|---|---|---|---|---|---|---|---|---|---|---|---|---|---|
| **#Points** | | | | | | | | | | | | | | |
| **Noise Level** | | | 1% Noise | | 2% Noise | | 3% Noise | | 1% Noise | | 2% Noise | | 3% Noise | |
| **Type** | **Method** | **Venue** | *CD↓* | *P2M↓* | *CD↓* | *P2M↓* | *CD↓* | *P2M↓* | *CD↓* | *P2M↓* | *CD↓* | *P2M↓* | *CD↓* | *P2M↓* |
| Optim. | Jet (Cazals & Pouget, 2005) | CAGD 05 | 27.12 | 6.13 | 41.55 | 13.47 | 62.62 | 29.21 | 8.51 | 2.07 | 24.32 | 14.03 | 57.88 | 42.67 |
| | Bilateral (Digne & De Franchis, 2017) | IPL 17 | 36.46 | 13.42 | 50.07 | 20.18 | 69.98 | 35.57 | 8.77 | 2.34 | 23.76 | 13.89 | 63.04 | 47.30 |
| | MRPCA (Mattei & Castrodad, 2017) | CGF 17 | 29.72 | 9.22 | 37.28 | 11.17 | 50.09 | 19.63 | 6.69 | 0.99 | 20.08 | 10.03 | 57.75 | 40.81 |
| | GLR (Zeng et al., 2019) | TIP 19 | 29.59 | 10.52 | 37.73 | 13.06 | 49.09 | 21.14 | 6.96 | 1.61 | 15.87 | 8.30 | 38.39 | 27.07 |
| | Lowrank (Lu et al., 2020) | TVCG 20 | 36.89 | 12.87 | 79.38 | 43.70 | 133.82 | 90.28 | 18.69 | 9.99 | 50.48 | 37.58 | 98.35 | 81.14 |
| Sup. | DeepPSR (Chen et al., 2022a) | TPAMI 22 | 23.53 | 3.06 | 33.50 | 7.34 | 40.72 | 12.38 | 6.49 | 0.76 | 9.97 | 2.96 | 13.44 | 5.31 |
| | IPFN (de Silva Edirimuni et al., 2023b) | CVPR 23 | 20.56 | 2.18 | 30.43 | 5.55 | 42.41 | 13.76 | 6.05 | 0.59 | 8.03 | 1.82 | 19.71 | 10.12 |
| | PathNet (Wei et al., 2024) | TPAMI 24 | 26.72 | 5.84 | 39.73 | 12.99 | 45.24 | 24.04 | 7.16 | 1.24 | 11.40 | 4.10 | 18.75 | 9.52 |
| | PD-Refiner (Zhang et al., 2024) | ACM MM 24 | 17.54 | 1.66 | 24.44 | 4.49 | 30.77 | 9.13 | 4.66 | 0.45 | 6.53 | 1.64 | 12.28 | 5.75 |
| | StraightPCF (de Silva Edirimuni et al., 2024) | CVPR 24 | 18.70 | 2.39 | 26.44 | 6.04 | 32.87 | 11.26 | 5.62 | 1.11 | 7.65 | 2.66 | 13.07 | 6.48 |
| | PD-LTS (Mao et al., 2024) | CVPR 24 | 17.81 | 1.82 | 24.41 | 4.70 | 34.33 | 11.98 | 4.70 | 0.54 | 6.46 | 1.82 | 18.52 | 10.67 |
| | ASDN (Guo et al., 2025) | AAAI 25 | 18.80 | 2.19 | 25.81 | 5.15 | 30.79 | 9.46 | 5.15 | 0.69 | 6.76 | 1.88 | 13.04 | 6.55 |
| | HybridPF (Edirimuni et al., 2025) | Arxiv 25 | 18.30 | 2.10 | 25.50 | 5.20 | 32.30 | 11.10 | 4.90 | 0.60 | 7.00 | 2.00 | 12.70 | 6.10 |
| Unsup. | TotalDn (Hermosilla et al., 2019) | ICCV 19 | 33.90 | 8.26 | 72.51 | 34.85 | 133.85 | 87.40 | 10.24 | 3.14 | 27.22 | 15.67 | 74.74 | 57.29 |
| | DMR-U (Luo & Hu, 2020) | ACM MM 20 | 53.13 | 25.22 | 64.55 | 33.17 | 81.34 | 46.47 | 12.26 | 5.21 | 21.38 | 12.51 | 24.96 | 15.20 |
| | Score-U (Luo & Hu, 2021) | ICCV 21 | 31.07 | 8.88 | 46.75 | 18.29 | 72.25 | 37.62 | 9.18 | 2.65 | 24.39 | 14.11 | 53.03 | 38.41 |
| | Noise2Score3D (Wei et al., 2025) | ICCV 25 | 28.48 | 11.06 | 41.90 | 18.18 | 55.83 | 29.47 | 8.33 | 4.61 | 15.32 | 9.70 | 24.34 | 17.04 |
| | U-CAN (Zhou et al., 2025a) | NeurIPS 25 | 24.97 | 11.05 | 32.34 | 12.55 | 36.66 | 18.42 | 8.35 | 6.09 | 9.75 | 6.75 | 24.79 | 18.63 |
| | SIMPC (Ours) | | 20.25 | 3.60 | 28.82 | 7.13 | 34.42 | 13.85 | 5.81 | 1.02 | 8.33 | 3.38 | 12.58 | 6.45 |

*Table 1.* Denoising results of different methods on PUNet (Yu et al., 2018) under Gaussian noise. Both CD and P2M distances are multiplied by $10^5$. For each column, the top-3 results among unsupervised methods are highlighted with **1st.** , 2nd. , and 3rd. .

| | | | 10K Points | | | | | | 50K Points | | | | | |
|---|---|---|---|---|---|---|---|---|---|---|---|---|---|---|
| **#Points** | | | | | | | | | | | | | | |
| **Noise Level** | | | 1% Noise | | 2% Noise | | 3% Noise | | 1% Noise | | 2% Noise | | 3% Noise | |
| **Type** | **Method** | **Venue** | *CD↓* | *P2M↓* | *CD↓* | *P2M↓* | *CD↓* | *P2M↓* | *CD↓* | *P2M↓* | *CD↓* | *P2M↓* | *CD↓* | *P2M↓* |
| Optim. | Jet (Cazals & Pouget, 2005) | CAGD 05 | 30.32 | 8.30 | 52.98 | 13.72 | 76.50 | 22.27 | 10.91 | 1.80 | 25.82 | 7.00 | 57.87 | 21.44 |
| | Bilateral (Digne & De Franchis, 2017) | IPL 17 | 43.20 | 13.51 | 61.71 | 16.46 | 82.95 | 23.92 | 11.72 | 1.98 | 24.78 | 6.34 | 60.77 | 21.89 |
| | MRPCA (Mattei & Castrodad, 2017) | CGF 17 | 33.23 | 9.31 | 48.74 | 11.78 | 65.02 | 16.76 | 9.66 | 1.40 | 21.53 | 4.78 | 55.70 | 19.76 |
| | GLR (Zeng et al., 2019) | TIP 19 | 33.99 | 9.56 | 52.74 | 11.46 | 72.49 | 16.74 | 9.64 | 1.34 | 20.15 | 4.17 | 44.88 | 13.06 |
| Sup. | DeepPSR (Chen et al., 2022a) | TPAMI 22 | 28.73 | 7.83 | 47.57 | 11.18 | 60.31 | 16.19 | 10.10 | 1.46 | 15.15 | 3.40 | 20.93 | 5.73 |
| | IterativePFN (de Silva Edirimuni et al., 2023b) | CVPR 23 | 26.20 | 7.00 | 44.40 | 10.10 | 60.30 | 15.60 | 9.10 | 1.40 | 12.50 | 2.40 | 25.30 | 7.20 |
| | PathNet (Wei et al., 2024) | TPAMI 24 | 30.03 | 8.65 | 41.05 | 14.22 | 56.63 | 22.39 | 9.91 | 1.48 | 14.92 | 3.15 | 23.10 | 5.95 |
| | PD-Refiner (Zhang et al., 2024) | ACM MM 24 | 28.26 | 5.69 | 40.09 | 8.68 | 49.33 | 12.05 | 8.18 | 1.04 | 11.71 | 2.20 | 19.95 | 4.84 |
| | StraightPCF (de Silva Edirimuni et al., 2024) | CVPR 24 | 27.50 | 5.40 | 40.50 | 7.90 | 49.20 | 10.90 | 8.80 | 1.40 | 11.70 | 2.60 | 18.20 | 4.50 |
| | PD-LTS (Mao et al., 2024) | CVPR 24 | 28.40 | 5.40 | 41.50 | 8.30 | 53.40 | 12.60 | 8.40 | 1.30 | 12.10 | 2.50 | 20.20 | 5.30 |
| | HybridPF (Edirimuni et al., 2025) | Arxiv 25 | 27.90 | 4.70 | 39.70 | 7.00 | 49.90 | 11.20 | 7.90 | 1.00 | 11.20 | 2.00 | 19.10 | 4.80 |
| Unsup. | DMR-U (Luo & Hu, 2020) | ACM MM 20 | 94.61 | 45.84 | 108.77 | 51.39 | 126.39 | 58.02 | 21.85 | 7.46 | 33.18 | 11.99 | 56.16 | 22.07 |
| | Score-U (Luo & Hu, 2021) | ICCV 21 | 42.32 | 12.61 | 60.34 | 17.63 | 103.02 | 43.79 | 12.34 | 2.22 | 24.92 | 6.60 | 39.28 | 11.74 |
| | Noise2Score3D (Wei et al., 2025) | ICCV 25 | 35.23 | 10.37 | 67.30 | 19.64 | 93.04 | 31.50 | 15.75 | 3.55 | 31.32 | 9.42 | 48.83 | 18.22 |
| | SIMPC (Ours) | | 32.10 | 6.57 | 43.18 | 8.69 | 53.33 | 12.48 | 8.86 | 1.07 | 13.40 | 2.41 | 18.62 | 4.15 |

*Table 2.* Denoising results of different methods on PCNet (Rakotosaona et al., 2020) under Gaussian noise. Both CD and P2M distances are multiplied by $10^5$. For each column, the top-3 results among unsupervised methods are highlighted with **1st.** , 2nd. , and 3rd. .

NVIDIA GeForce RTX 4090 GPU. We train the network using the Adam optimizer with a learning rate of $1 \times 10^{-4}$ for 100 epochs and a batch size of 16. We evaluate denoising performance using Chamfer Distance (CD) and Point-to-Mesh (P2M) distance (Ravi et al., 2020).

**Compared Methods.** We compare SIMPC with representative optimization-based methods (e.g., GLR (Zeng et al., 2019)), supervised methods (e.g., PD-Refiner (Zhang et al., 2024) and StraightPCF (de Silva Edirimuni et al., 2024)), and unsupervised methods (e.g., Noise2Score3D (Wei et al., 2025) and U-CAN (Zhou et al., 2025a)).

### 5.2. Experiments on Synthetic Noise

Quantitative results on Gaussian noise for PUNet and PCNet are reported in Tab. 1 and Tab. 2, while results on four non-Gaussian noise for PUNet are presented in Tab. 3 and Tab. 7 (Appendix). Correspondingly, qualitative visualizations on both datasets are shown in Fig. 4 and Fig. 7 (Appendix).

Optimization-based methods generally exhibit limited denoising capability and unstable performance across different noise intensities and resolutions. By contrast, supervised methods achieve stronger geometric fidelity by learning direct mappings toward the clean surface under explicit supervision. Among unsupervised methods, density-based approaches such as TotalDen, DMR-U, and Score-U often produce sub-optimal denoising results with noticeable residual noise, and their iterative processes tend to succumb to clustering artifacts. Meanwhile, statistical mapping-based methods such as Noise2Score3D and U-CAN exhibit relatively high P2M errors under comparable CD values, indicating that the denoised points are not well aligned with the target surface. In contrast, our proposed SIMPC establishes deterministic correspondences between noisy variants associated with the same target surface. As a result, it achieves superior performance over existing unsupervised methods, with more precise target surface alignment and substantially reduced residual noise.

| #Points | | 10K Points | | | | | | 50K Points | | | | | |
|---|---|---|---|---|---|---|---|---|---|---|---|---|---|
| Noise Level | | 1% noise | | 2% noise | | 3% noise | | 1% noise | | 2% noise | | 3% noise | |
| Type | Methods | CD↓ | P2M↓ | CD↓ | P2M↓ | CD↓ | P2M↓ | CD↓ | P2M↓ | CD↓ | P2M↓ | CD↓ | P2M↓ |
| Laplacian Sup. | PD-Refiner (Zhang et al., 2024) | 20.62 | 2.66 | 28.96 | 7.94 | 42.48 | 17.93 | 5.28 | 0.85 | 8.89 | 3.16 | 21.86 | 13.33 |
| | PD-LTS (Mao et al., 2024) | 20.85 | 2.84 | 28.67 | 8.25 | 51.77 | 26.55 | 5.31 | 0.96 | 9.80 | 4.28 | 44.76 | 34.71 |
| | StraightPCF (de Silva Edirimuni et al., 2024) | 21.86 | 3.44 | 30.38 | 9.45 | 47.98 | 23.75 | 6.07 | 1.39 | 9.81 | 4.32 | 27.95 | 19.61 |
| Laplacian Unsup. | Score-U (Luo & Hu, 2021) | 36.94 | 11.87 | 61.69 | 29.88 | 98.92 | 58.90 | 12.41 | 4.76 | 35.13 | 23.63 | 60.30 | 43.36 |
| | Noise2Score3D (Wei et al., 2025) | 43.08 | 17.43 | 80.66 | 46.86 | 125.06 | 84.68 | 20.33 | 11.31 | 48.47 | 35.63 | 85.73 | 70.64 |
| | SIMPC (Ours) | 23.65 | 4.78 | 33.37 | 10.98 | 41.91 | 19.59 | 6.81 | 1.78 | 10.07 | 4.48 | 25.16 | 16.72 |
| Discrete Sup. | PD-Refiner (Zhang et al., 2024) | 6.13 | 0.75 | 15.34 | 1.50 | 19.49 | 2.56 | 3.24 | 0.12 | 4.25 | 0.37 | 5.12 | 0.83 |
| | PD-LTS (Mao et al., 2024) | 6.45 | 0.86 | 15.63 | 1.58 | 19.85 | 2.76 | 3.37 | 0.16 | 4.29 | 0.40 | 5.15 | 0.90 |
| | StraightPCF (de Silva Edirimuni et al., 2024) | 6.55 | 1.03 | 16.82 | 2.50 | 21.04 | 3.97 | 3.86 | 0.50 | 5.32 | 1.30 | 6.01 | 1.80 |
| Discrete Unsup. | Score-U (Luo & Hu, 2021) | 23.44 | 7.57 | 29.79 | 8.71 | 57.72 | 29.39 | 5.79 | 0.82 | 8.56 | 2.21 | 14.53 | 5.07 |
| | Noise2Score3D (Wei et al., 2025) | 11.84 | 3.02 | 28.33 | 8.96 | 39.10 | 16.71 | 6.49 | 1.96 | 12.25 | 6.37 | 19.78 | 12.36 |
| | SIMPC (Ours) | 8.02 | 1.89 | 17.83 | 3.21 | 22.27 | 4.72 | 3.74 | 0.32 | 5.08 | 0.82 | 6.34 | 1.57 |

*Table 3.* Denoising results of different methods on PUNet (Yu et al., 2018) under non-Gaussian noise. Both CD and P2M distances are multiplied by $10^5$. For each column, the top-3 results among unsupervised methods are highlighted with **1st.**, 2nd., and 3rd..

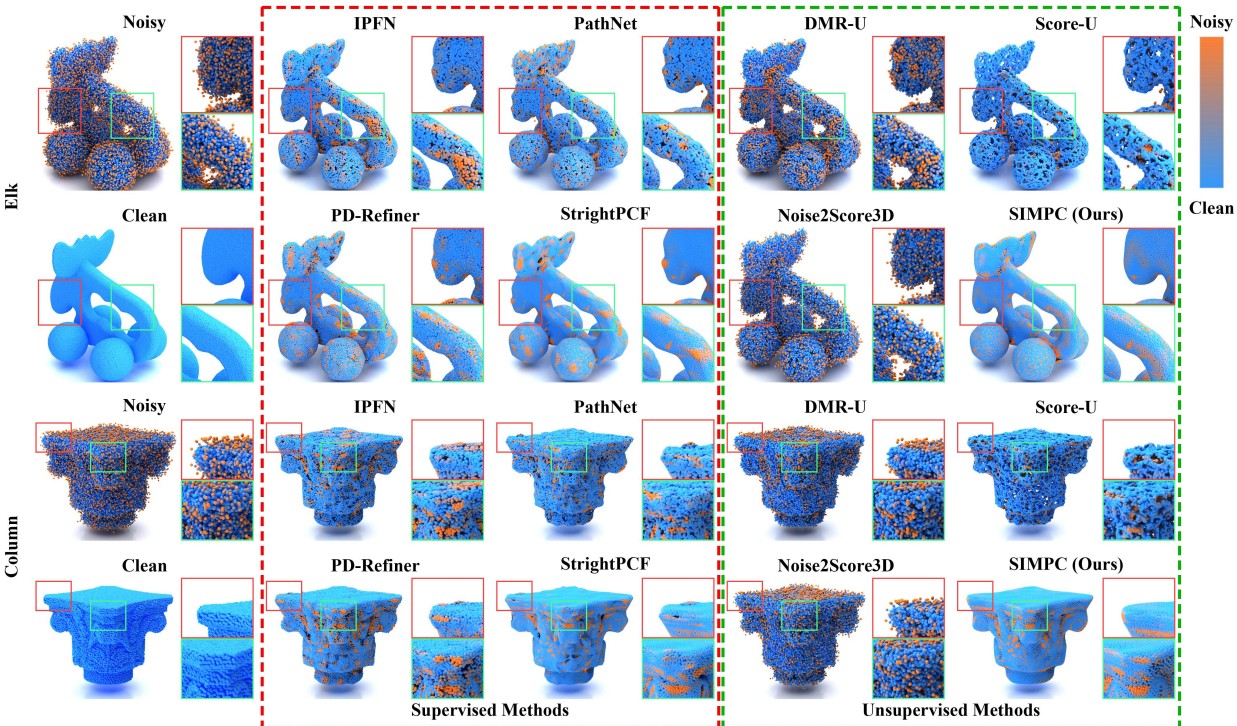

*Figure 4.* Visual results for 50K-resolution shapes from the PCNet and PUNet datasets under 3% Gaussian noise relative to the bounding sphere radius. Point colors represent the point-wise P2M distance, where orange indicates larger errors and blue denotes cleaner regions.

## 5.3. Experiments on Real-World Noise

Qualitative results on the Paris-Rue-Madame dataset are shown in Fig. 5, and those on the Kinect dataset are shown in Fig. 6 and Fig. 8 (Appendix). Quantitative results on the Kinect dataset are reported in Tab. 5.

Overall, supervised methods exhibit stable denoising on real-world scans, producing smooth surfaces. Among unsupervised methods, DMR-U leaves noticeable residual noise and surface discontinuities, while Score-U often converges to linear structures, leading to severe clustering artifacts. Meanwhile, Noise2Score3D provide only marginal noise suppres-

sion and fail to accurately align noisy points to the underlying surface. In contrast, our proposed SIMPC demonstrates satisfactory generalization on real-world datasets in both quantitative and qualitative evaluations.

## 5.4. Ablation Study

We further conduct ablation studies on the loss design and the extent of denoising vector extension, as shown in Fig. 5.

**Loss Type.** We first investigate the effect of different loss design strategies. When only the Chamfer Distance $\mathcal{L}_{SR(CD)}$ is used to provide a coarse similarity constraint, the denois-

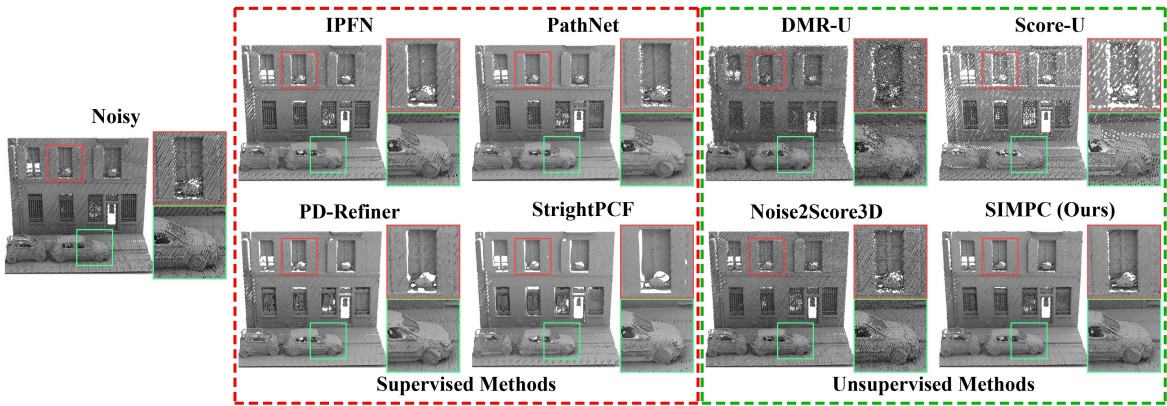

*Figure 5.* Visual results for a real scan and zoom-in views of several objects from the Paris-Rue-Madame dataset (Serna et al., 2014). Supervised and unsupervised methods, including our proposed SIMPC, are highlighted in red and green, respectively.

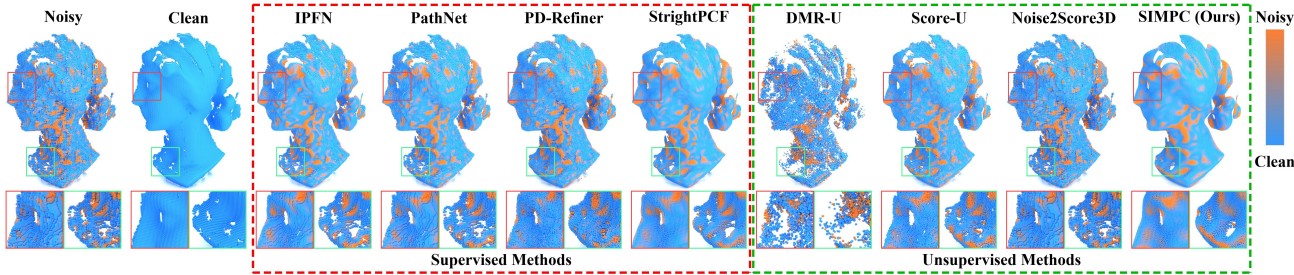

*Figure 6.* Visual results for a real scan and zoom-in views of several surface regions from the Kinect dataset (Wang et al., 2016). Supervised and unsupervised methods, including our proposed SIMPC, are highlighted in red and green, respectively.

| Type | Method | CD↓ | P2M↓ |
|------|--------|-----|------|
| Sup. | IPFN (de Silva Edirimuni et al., 2023b) | 14.92 | 7.99 |
| | PathNet (Wei et al., 2024) | 14.20 | 7.38 |
| | PD-Refiner (Zhang et al., 2024) | 14.34 | 7.61 |
| | StrightPCF (de Silva Edirimuni et al., 2024) | 13.46 | 7.39 |
| Unsup. | DMR-U (Luo & Hu, 2020) | 45.18 | 23.38 |
| | Score-U (Luo & Hu, 2021) | 15.85 | 7.33 |
| | Noise2Score3D (Wei et al., 2025) | 15.25 | 8.11 |
| | SIMPC (Ours) | **13.01** | **6.35** |

*Table 4.* Denoising comparison in Kinect dataset (Wang et al., 2016). For each column, the top-3 results among unsupervised methods are highlighted with **1st.** , 2nd. , and 3rd. .

| #Points | | 10K Points | | | | | |
|---------|--|-----------|--|--|--|--|--|
| **Noise Level** | | 1% Noise | | 2% Noise | | 3% Noise | |
| **Ablation Settings** | | CD↓ | P2M↓ | CD↓ | P2M↓ | CD↓ | P2M↓ |
| **Loss Type** | $\mathcal{L}_{SR(CD)}$ | 18.91 | 2.47 | 37.84 | 13.91 | 65.73 | 36.20 |
| | $\mathcal{L}_{SR(EMD)}$ | 26.54 | 7.64 | 31.88 | 12.41 | 41.04 | 18.87 |
| | $\mathcal{L}_{MPC} + \mathcal{L}_{SR(CD)}$ | 20.25 | 3.60 | 28.82 | 7.13 | 34.42 | 13.85 |
| **Extend Dist** | **Near** ($w_2 = 1.5$) | 20.77 | 4.06 | 29.31 | 7.68 | 35.10 | 14.55 |
| | **Symmetry** ($w_2 = 2$) | 20.25 | 3.60 | 28.82 | 7.13 | 34.42 | 13.85 |
| | **Far** ($w_2 = 2.5$) | 21.39 | 4.84 | 30.13 | 8.54 | 36.33 | 15.25 |

*Table 5.* Denoising results of different ablation settings on PUNet (Yu et al., 2018) under Gaussian noise. Both CD and P2M distances are multiplied by $10^5$.

ing performance drops significantly as the noise intensity increases. Introducing the Earth Mover's Distance $\mathcal{L}_{SR(EMD)}$ to impose an ambiguous one-to-one correspondence improves the denoising capability compared with $\mathcal{L}_{SR(CD)}$, yet still suffers from inaccurate surface localization. In contrast, our proposed $\mathcal{L}_{MPC}$ establishes deterministic surface correspondences, enabling accurate surface localization and achieving the best denoising performance, highlighting the necessity of learning deterministic correspondences for effective unsupervised denoising.

**Extension Distance.** Furthermore, we study the influence of different scaling factors $w_2$ that control the extent of denoising vector extension for mirror-point generation. The results show that using the geometric symmetry design yields the better performance. By contrast, the other non-symmetry settings lead to sub-optimal results due to residual perturbations during denoising training.

## 6. Conclusion

We propose Self-Induced Mirror-Point Consistency (SIMPC) to learn deterministic correspondences between points and the underlying surface in an unsupervised manner. The key insight of SIMPC is to establish correspondences between points associated with the same target surface by leveraging geometric priors extracted during the denoising process, rather than matching variants across independently sampled noisy observations. With such deterministic correspondences, SIMPC is able to more accurately localize the underlying surface from noisy point clouds. We evaluate SIMPC on both synthetic datasets and real-world scanned data under various noise conditions. Extensive experimental results demonstrate that SIMPC consistently achieves state-of-the-art performance among unsupervised methods and even surpasses several strong supervised counterparts.

## Impact Statement

This paper presents work whose goal is to advance the field of machine learning. There are many potential societal consequences of our work, none of which we feel must be specifically highlighted here.

## Acknowledgments

This work is supported by the National Key Research and Development Program of China (Distributed Polarization 3D Imaging Radar System Technology, No. 2022YFB3901601).

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

# A. Appendix

## A.1. Runtimes and Parameters

We compare SIMPC with previous state-of-the-art point cloud denoising methods in terms of runtime and the number of parameters. The results are reported in Tab. 6. Overall, SIMPC achieves comparable runtime and parameter complexity to existing methods.

| Type | Method | Time (s)↓ | Para (M)↓ |
|------|--------|-----------|-----------|
| Sup. | **IPFN** (de Silva Edirimuni et al., 2023b) | 4.89 | 3.20 |
| | **PathNet** (Wei et al., 2024) | 11.10 | 15.61 |
| | **PD-Refiner** (Zhang et al., 2024) | 1.41 | 15.11 |
| | **StrightPCF** (de Silva Edirimuni et al., 2024) | 1.46 | 0.53 |
| Unsup. | **DMR-U** (Luo & Hu, 2020) | 0.43 | 0.23 |
| | **Score-U** (Luo & Hu, 2021) | 1.18 | 0.19 |
| | **Noise2Score3D** (Wei et al., 2025) | 0.14 | 24.38 |
| | **SIMPC (Ours)** | 1.01 | 0.63 |

*Table 6.* Runtimes of state-of-the-art methods on one object with 10K points and 2% Gaussian noise, from the PUNet dataset.

## A.2. Additional Results on Non-Gaussian Noise

We further report quantitative comparisons on the PUNet dataset under two additional non-Gaussian synthetic noise types, namely Anisotropic and Uniform noise. The results are summarized in Tab. 7. SIMPC achieves satisfactory generalization across different noise distributions.

| #Points | | | 10K Points | | | | | | 50K Points | | | | | |
|---------|--|--|------------|--|--|--|--|--|------------|--|--|--|--|--|
| Noise Level | | | 1% noise | | 2% noise | | 3% noise | | 1% noise | | 2% noise | | 3% noise | |
| Type | | Methods | CD↓ | P2M↓ | CD↓ | P2M↓ | CD↓ | P2M↓ | CD↓ | P2M↓ | CD↓ | P2M↓ | CD↓ | P2M↓ |
| Anisotropic | Sup. | **PD-Refiner** (Zhang et al., 2024) | 17.41 | 1.76 | 24.70 | 4.80 | 33.43 | 11.28 | 4.66 | 0.45 | 6.97 | 1.98 | 18.20 | 10.42 |
| | | **PD-LTS** (Mao et al., 2024) | 17.62 | 1.85 | 24.91 | 5.07 | 40.52 | 16.78 | 4.70 | 0.52 | 7.04 | 2.19 | 25.18 | 16.08 |
| | | **StraightPCF** (de Silva Edirimuni et al., 2024) | 18.54 | 2.50 | 26.43 | 6.21 | 36.77 | 14.37 | 5.62 | 1.12 | 8.06 | 3.06 | 19.16 | 11.32 |
| | Unsup. | **Score-U** (Luo & Hu, 2021) | 32.02 | 9.34 | 47.58 | 18.40 | 81.73 | 45.02 | 9.44 | 2.53 | 23.26 | 13.01 | 37.79 | 23.11 |
| | | **Noise2Score3D** (Wei et al., 2025) | 32.77 | 10.46 | 58.14 | 27.17 | 81.87 | 45.87 | 13.87 | 6.16 | 29.30 | 18.55 | 48.12 | 35.29 |
| | | **SIMPC (Ours)** | 20.12 | 3.68 | 29.35 | 7.57 | 35.39 | 14.28 | 5.77 | 1.13 | 8.52 | 3.56 | 18.84 | 11.44 |
| Uniform | Sup. | **PD-Refiner** (Zhang et al., 2024) | 6.12 | 0.76 | 17.11 | 1.52 | 21.65 | 2.62 | 3.63 | 0.12 | 4.54 | 0.37 | 5.34 | 0.89 |
| | | **PD-LTS** (Mao et al., 2024) | 6.43 | 0.84 | 17.34 | 1.59 | 21.83 | 2.74 | 3.69 | 0.15 | 4.59 | 0.44 | 5.82 | 1.27 |
| | | **StraightPCF** (de Silva Edirimuni et al., 2024) | 6.57 | 1.04 | 18.51 | 2.59 | 23.78 | 4.46 | 4.21 | 0.51 | 5.66 | 1.37 | 6.97 | 2.40 |
| | Unsup. | **Score-U** (Luo & Hu, 2021) | 23.82 | 7.63 | 31.37 | 8.85 | 58.21 | 29.07 | 6.00 | 0.80 | 8.86 | 2.05 | 15.56 | 5.29 |
| | | **Noise2Score3D** (Wei et al., 2025) | 12.18 | 3.30 | 31.80 | 9.12 | 43.01 | 15.22 | 7.34 | 1.78 | 12.53 | 4.87 | 18.32 | 9.21 |
| | | **SIMPC (Ours)** | 8.10 | 1.92 | 19.78 | 3.43 | 25.02 | 4.94 | 4.10 | 0.32 | 5.56 | 0.86 | 7.06 | 1.92 |

*Table 7.* Denoising results of different methods on PUNet (Yu et al., 2018) under non-Gaussian noise. Both CD and P2M distances are multiplied by $10^5$. For each column, the top-3 results among unsupervised methods are highlighted with **1st.** , 2nd. , and 3rd. .

## A.3. Additional Visual Results

We also provide additional qualitative denoising results on more objects from the PUNet, PCNet, and Kinect datasets. The visual comparisons are shown in Fig. 7 and Fig. 8. As can be observed, our method achieves more effective residual noise suppression and produces smoother surfaces across different datasets.

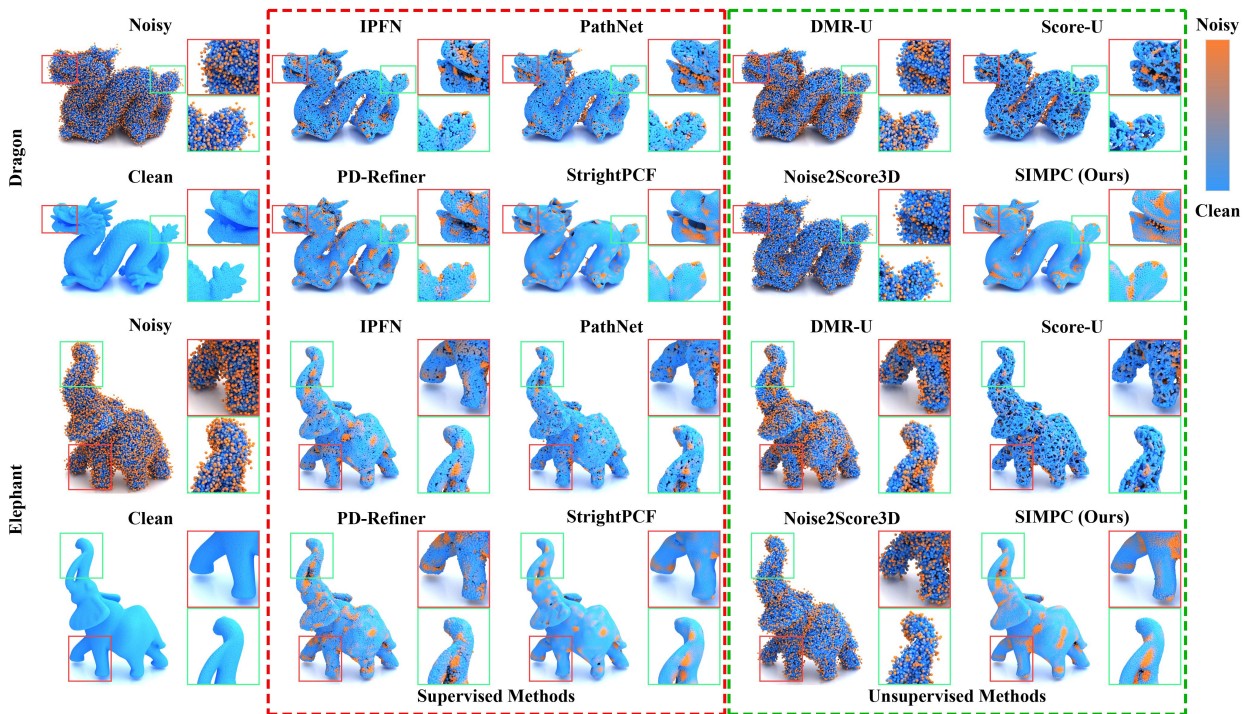

*Figure 7.* Visual results for 50K resolution shapes from the PCNet and PUNet dataset with 3% Gaussian noise on the bounding sphere radius. Point colors represent the point-wise P2M distance, where orange indicates larger errors and blue denotes cleaner regions.

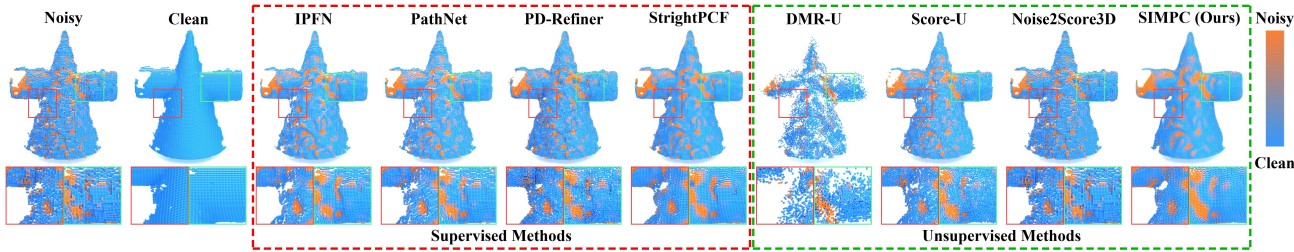

*Figure 8.* Visual results for a real scan and zoom-in views of several surface regions from the Kinect dataset (Wang et al., 2016). Supervised and unsupervised methods, including our proposed SIMPC, are highlighted in red and green, respectively.

## A.4. Limitation and Generalization Boundary

| #Points | | | 5K Points | | | |
|---|---|---|---|---|---|---|
| Noise Level | 1% Noise | | 2% Noise | | 3% Noise | |
| Methods | CD↓ | P2M↓ | CD↓ | P2M↓ | CD↓ | P2M↓ |
| Noise2Score3D (Wei et al., 2025) | 50.31 | 17.15 | 69.36 | 26.83 | 88.03 | 45.32 |
| SIMPC (Ours) | 42.85 | 11.89 | 58.04 | 15.77 | 68.06 | 21.18 |
| StraightPCF (de Silva Edirimuni et al., 2024) | 33.48 | 6.37 | 53.39 | 12.47 | 64.19 | 19.32 |

*Table 8.* Denoising results of extremely sparse point clouds settings on PUNet (Yu et al., 2018) under Gaussian noise. Both CD and P2M distances are multiplied by $10^5$.

The generalization boundaries of SIMPC mainly lie in a slight performance degradation when handling extremely sparse point clouds and a mild smoothing effect on sharp geometric structures. We evaluate SIMPC and other methods on extremely sparse point clouds, as shown in Tab. 8. Compared with the supervised method (StraightPCF (de Silva Edirimuni et al., 2024)), the performance gap of SIMPC slightly increases under this setting. Nevertheless, SIMPC still achieves clear improvements over previous unsupervised methods such as Noise2Score3D (Wei et al., 2025). In addition, SIMPC achieves performance comparable to supervised methods in flat regions. However, for sharp geometric structures, it exhibits slight smoothing compared to supervised approaches. This is mainly due to the absence of ground-truth geometric details, which are typically leveraged in supervised denoising frameworks.

## A.5. A Manifold-Based View of Mirror-Point Consistency

**Setup and notation.** Let $\mathcal{X} = \{x_i\}_{i=1}^N \subset \mathbb{R}^3$ be a given noisy point cloud (a single observation), where $N$ is the number of points and $x_i \in \mathbb{R}^3$ denotes the $i$-th noisy point. We model the underlying clean surface as a compact $C^2$ two-dimensional manifold $\mathcal{M} \subset \mathbb{R}^3$ (Luo & Hu, 2020). For any point $x \in \mathbb{R}^3$, we define its Euclidean distance to the surface as $\mathrm{dist}(x, \mathcal{M}) = \min_{y \in \mathcal{M}} \|x - y\|_2$. For each $x_i$ considered in the following analysis, its nearest-point projection onto $\mathcal{M}$ is produced by

$$g_i = \Pi(x_i) = \arg \min_{y \in \mathcal{M}} \|x_i - y\|_2, \tag{14}$$

where $\Pi(\cdot)$ denotes the closest-point projection operator, and $g_i \in \mathcal{M}$ is regarded as the ideal surface target associated with the noisy point $x_i$. When the closest point is unique and $\mathcal{M}$ is $C^2$, the displacement $x_i - g_i$ is orthogonal to the tangent plane $T_{g_i}\mathcal{M}$, i.e., $x_i - g_i \perp T_{g_i}\mathcal{M}$, and is therefore colinear with the surface normal at $g_i$. Let $n(g_i) \in \mathbb{R}^3$ be a unit normal vector of $\mathcal{M}$ at $g_i$, and let $T_{g_i}\mathcal{M}$ be the tangent plane at $g_i$, which specify the normal and tangential directions used to model local denoising uncertainty.

**Local neighborhoods and denoising updates.** For each point $x_i \in \mathcal{X}$, denote its $k$-NN index set by $\mathcal{N}_i = \mathrm{KNN}(x_i, \mathcal{X}, k)$. Based on the local neighborhood $\mathcal{N}_i$, the denoising process predicts a point-wise displacement (denoising vector) $d_i \in \mathbb{R}^3$ for $x_i$. The denoised seed point is then obtained by the residual update

$$\hat{x}_i = x_i + w_1 d_i, \qquad w_1 = 1. \tag{15}$$

Following MPGM, we further extend the displacement to generate a mirror input

$$\tilde{x}_i = x_i + w_2 d_i, \qquad w_2 > w_1, \tag{16}$$

and recompute its neighborhood $\tilde{\mathcal{N}}_i = \mathrm{KNN}(\tilde{x}_i, \mathcal{X} \setminus \{x_i\}, k)$. Intuitively, since $\mathcal{L}_{\mathrm{SR}}$ encourages $d_i$ to point toward the local surface, extrapolating with a larger step $w_2 > w_1$ tends to move $\tilde{x}_i$ beyond the local surface patch, yielding a shifted (often complementary) neighborhood for the second denoising pass. Applying the same denoising procedure at $\tilde{x}_i$ under $\tilde{\mathcal{N}}_i$ yields a mirror displacement $\tilde{d}_i \in \mathbb{R}^3$ and the denoised mirror point

$$\bar{x}_i = \tilde{x}_i + \tilde{d}_i. \tag{17}$$

Consequently, each noisy seed $x_i$ deterministically induces a paired output $(\hat{x}_i, \bar{x}_i)$.

**Statistical model of the denoising landing region.** We model the *landing position* of each denoised point as a random variable whose distribution captures residual bias and uncertainty induced by local geometry and neighborhood variations. For a given $x_i$, we assume the seed landing $\hat{x}_i$ follows an anisotropic Gaussian distribution whose covariance is structured along the local tangential and normal directions at the associated surface location $g_i$. Concretely, let $\hat{n}_i = n(g_i)$ and $\hat{T}_i = T_{g_i}\mathcal{M}$, then

$$\hat{x}_i \sim \mathcal{N}\big(\hat{\mu}_i, \hat{\Sigma}_{i,\|} + \hat{\sigma}_{i,\perp}^2 \hat{n}_i\hat{n}_i^\top\big), \tag{18}$$

where $\hat{\mu}_i \in \mathbb{R}^3$ is the mean landing position, $\hat{\Sigma}_{i,\|} \succeq 0$ models the tangential uncertainty (supported on the tangent subspace $\hat{T}_i$), and $\hat{\sigma}_{i,\perp}^2$ models the normal uncertainty along $\hat{n}_i$. Since the $\mathcal{L}_{\mathrm{SR}}$ term provides a coarse-grained similarity regularization that encourages denoised outputs to move toward the underlying surface, the seed mean landing position is closer to $\mathcal{M}$ than the noisy observation, i.e.,

$$\mathrm{dist}(\hat{\mu}_i, \mathcal{M}) \leq \mathrm{dist}(x_i, \mathcal{M}). \tag{19}$$

The mirror landing $\bar{x}_i$ follows the same distributional form as in Eq. (18), but with branch-specific parameters $\bar{\mu}_i \in \mathbb{R}^3$, $\bar{\Sigma}_{i,\|} \succeq 0$, and $\bar{\sigma}_{i,\perp}^2$, defined under the same local frame $\bar{n}_i = n(g_i)$ and $\bar{T}_i = T_{g_i}\mathcal{M}$, which capture the mean landing position and the tangential/normal uncertainties induced by the mirror neighborhood.

**From displacement extension to a coupled sequence of landing distributions.** Starting from a noisy seed point $x_i$, MPGM induces a *coupled* sequence of (partly stochastic) landings through displacement extension and neighborhood re-sampling. For clarity, we summarize the induced transition as the following chain:

$$x_i \xrightarrow{\text{denoise with } w_1} \hat{x}_i \sim \mathcal{N}\big(\hat{\mu}_i, \hat{\Sigma}_i\big) \xrightarrow{\text{extend } (w_2 > w_1)} \tilde{x}_i = x_i + w_2 d_i \xrightarrow{\text{denoise under } \tilde{\mathcal{N}}_i} \bar{x}_i \sim \mathcal{N}\big(\bar{\mu}_i, \bar{\Sigma}_i\big), \tag{20}$$

where $\hat{\Sigma}_i = \hat{\Sigma}_{i,\|} + \hat{\sigma}_{i,\perp}^2 \hat{n}_i\hat{n}_i^\top$ and $\bar{\Sigma}_i = \bar{\Sigma}_{i,\|} + \bar{\sigma}_{i,\perp}^2 \bar{n}_i\bar{n}_i^\top$ follow the structured form introduced in Eq. (18) (and implicitly depend on the neighborhood and the scaling factors, e.g., $w_1$). The intermediate point $\tilde{x}_i$ is generated from $(x_i, d_i)$, while its neighborhood is recomputed at $\tilde{x}_i$, which makes the subsequent mirror landing $\bar{x}_i$ exhibit branch-specific statistics. In particular, $\hat{x}_i$ and $\bar{x}_i$ form a *paired* output induced from the same seed $x_i$ via the shared displacement $d_i$.

Under the geometric setup, $x_i$ admits an associated ideal surface target $g_i = \Pi(x_i) \in \mathcal{M}$. With a mild locality condition (i.e., the mirror construction stays within the same local surface patch associated with $g_i$), it is natural to view the seed and mirror branches as producing two stochastic estimators of a *shared* surface-consistent target in the neighborhood of $g_i$, while allowing different biases and uncertainties due to the neighborhood change induced by $\tilde{x}_i$. Therefore, MPCL does not attempt to align two point sets globally; instead, it leverages the coupled chain in Eq. (20) to extract point-wise information about the shared surface target by directly reducing the discrepancy between the paired terminal samples $(\hat{x}_i, \bar{x}_i)$.

**MPCL as distribution alignment, variance contraction, and surface-target consistency.** Our mirror-point consistency loss (MPCL) enforces point-wise consistency between the paired landings:

$$\mathcal{L}_{\mathrm{MPC}} = \sum_{i=1}^{N} \big\|\hat{x}_i - \bar{x}_i\big\|_2^2. \tag{21}$$

Define the random difference $\Delta_i = \hat{x}_i - \bar{x}_i$. Conditioned on the same seed $x_i$, a standard second-moment expansion yields

$$\mathbb{E}\big[\|\Delta_i\|_2^2 \mid x_i\big] = \big\|\hat{\mu}_i - \bar{\mu}_i\big\|_2^2 + \mathrm{Tr}\big(\hat{\Sigma}_i + \bar{\Sigma}_i - 2\Sigma_{i,\times}\big), \tag{22}$$

where $\hat{\Sigma}_i = \hat{\Sigma}_{i,\|} + \hat{\sigma}_{i,\perp}^2 \hat{n}_i\hat{n}_i^\top$ and $\bar{\Sigma}_i = \bar{\Sigma}_{i,\|} + \bar{\sigma}_{i,\perp}^2 \bar{n}_i\bar{n}_i^\top$ denote the full covariances of the seed and mirror landings, respectively, and $\Sigma_{i,\times} = \mathrm{Cov}(\hat{x}_i, \bar{x}_i \mid x_i)$ is their cross-covariance. Eq. (22) shows that minimizing MPCL simultaneously (i) reduces the discrepancy between the landing means via $\|\hat{\mu}_i - \bar{\mu}_i\|_2^2$, and (ii) suppresses the combined uncertainty through the covariance trace term. Since the mirror branch is evaluated under a changed neighborhood around $\tilde{x}_i$, the two landings are generally not perfectly correlated, which limits $\Sigma_{i,\times}$ and makes MPCL effective in controlling $\mathrm{Tr}(\hat{\Sigma}_i + \bar{\Sigma}_i)$.

To connect this alignment to surface localization, recall that $g_i = \Pi(x_i)$ represents the ideal surface target associated with $x_i$. Under the landing model and the coarse-grained similarity regularization (Eq. (19)), both $\hat{\mu}_i$ and $\bar{\mu}_i$ are encouraged to lie in a local region closer to $\mathcal{M}$ than $x_i$. Then the mean-alignment term $\|\hat{\mu}_i - \bar{\mu}_i\|_2^2$ promotes a *shared* surface-consistent estimate within the same local patch around $g_i$: when two stochastic estimators tied to the same underlying target are forced

to consistent, the consistent point is constrained to remain near $\mathcal{M}$ (due to the similarity regularization), while reducing the ambiguity of where on $\mathcal{M}$ the point should land. Meanwhile, the uncertainty suppression term contracts the landing regions, making the paired outputs concentrate more tightly around the shared surface-consistent location. Consequently, MPGM provides a point-wise pairing $(\hat{x}_i, \bar{x}_i)$ for each noisy seed $x_i$, and MPCL enforces their consistency, thereby establishing a consistent relation between two variants that correspond to the same underlying surface target and facilitating more accurate localization of $g_i$ through distribution alignment and variance contraction.

### A.6. Deterministic and Geometric Symmetry Design of Mirror-Point Consistency

we further discuess the deterministic and geometric symmetry design by the analytical framework of SIMPC.

**Case (1): Symmetry and deterministic correspondence for GT surface**

For a noisy seed point $x_i$, let $s_i \in \mathcal{S}$ denote its corresponding target point on the underlying clean surface $\mathcal{S}$, and let the noise be written as $x_i = s_i + n_i$. At the current denoising stage, let the denoiser output a point-wise denoising vector $d_i = \mathcal{D}(x_i)$, so that the normally denoised seed point is $\hat{x}_i = x_i + d_i$. We further construct a mirror-point by extending the denoising vector $\tilde{x}_i = x_i + 2d_i$, and denote its denoised mirror-point by $\bar{x}_i = \tilde{x}_i + \tilde{d}_i$.

The Mirror-Point Consistency Loss is then

$$\mathcal{L}_{\mathrm{MPC}} = \mathbb{E}\left[|\hat{x}_i - \bar{x}_i|^2\right].$$

Through Taylor expansion, we can approximate the denoised seed point and the denoised mirror-point around the target surface point $s_i$:

$$\hat{x}_i = s_i + \frac{\partial \hat{x}_i}{\partial x_i} \cdot n_i + \mathcal{O}(n_i^2),$$

$$\bar{x}_i = s_i - \frac{\partial \hat{x}_i}{\partial x_i} \cdot n_i + \mathcal{O}(n_i^2).$$

Substituting the approximations, we get

$$\hat{x}_i - \bar{x}_i \approx 2 * \frac{\partial \hat{x}_i}{\partial x_i} \cdot n_i.$$

With further Taylor expansion, the loss function can be simplified as

$$\mathcal{L}_{\mathrm{MPC}} \approx 4 * \left|\frac{\partial \hat{x}_i}{\partial x_i} \cdot n_i\right|^2.$$

Through the consistency loss $\mathcal{L}_{\mathrm{MPC}}$, we optimize the network such that

$$\frac{\partial \hat{x}_i}{\partial x_i} \cdot n_i \to 0,$$

which implies that the denoising process becomes locally insensitive to symmetric perturbations around the same target surface point.

Through the consistency loss $\mathcal{L}_{\mathrm{MPC}}$, we optimize the network $\mathcal{D}$ such that $\frac{\partial \hat{x}_i}{\partial x_i} \cdot n_i$ is driven towards zero, which means that the output $\hat{x}_i$ tends to:

$$\hat{x}_i = \mathcal{D}(x_i) = \mathcal{D}(s_i + n_i)$$

$$= \mathcal{D}(s_i) + \frac{\partial \hat{x}_i}{\partial x_i} \cdot n_i + \mathcal{O}(n_i^2)$$

$$\approx \mathcal{D}(s_i).$$

Under the ideal assumption that the denoiser has learned the geometric prior of the underlying surface, we further have $\mathcal{D}(s_i) \approx s_i$.

Thus, the final denoised output satisfies $\hat{x}_i \approx s_i$, and similarly $\bar{x}_i \approx s_i$.

Therefore, the denoised seed point and its mirror counterpart converge to the same target surface point:

$$\hat{x}_i \approx \bar{x}_i \approx s_i \in \mathcal{S}.$$

In summary, the favorable convergence of SIMPC relies on two key conditions: deterministic correspondence and symmetry with respect to the same target surface. MPCL primarily suppresses the noise-sensitive component and drives both outputs toward a shared clean surface point. This behavior is consistent with the maximum-likelihood assumption in score-based denoising, where the ground-truth location is inferred unsupervised from noisy observations; under zero-mean Gaussian noise, this typically corresponds to the underlying clean surface.

**Case (2): Ambiguous correspondence with inconsistent denoising targets**

The derivation in Case (1) is based on the fact that the two points in the consistency pair correspond to the same underlying surface target. If the correspondence is ambiguous (constructed by noise-based or EMD-based methods), the two noisy points no longer share the same clean target. Let the paired points be associated with two different surface points $s_i^{(1)}$ and $s_i^{(2)}$ and independent noise condition, with

$$x_i^{(1)} = s_i^{(1)} + n_i^{(1)}, \qquad x_i^{(2)} = s_i^{(2)} + n_i^{(2)}, \qquad s_i^{(1)} \neq s_i^{(2)}.$$

Then their denoised outputs can be approximated as

$$\hat{x}_i^{(1)} = s_i^{(1)} + \frac{\partial \hat{x}_i^{(1)}}{\partial x_i^{(1)}} \cdot n_i^{(1)} + \mathcal{O}((n_i^{(1)})^2),$$

$$\hat{x}_i^{(2)} = s_i^{(2)} + \frac{\partial \hat{x}_i^{(2)}}{\partial x_i^{(2)}} \cdot n_i^{(2)} + \mathcal{O}((n_i^{(2)})^2).$$

Their difference becomes

$$\hat{x}_i^{(1)} - \hat{x}_i^{(2)} \approx (s_i^{(1)} - s_i^{(2)}) + \frac{\partial \hat{x}_i^{(1)}}{\partial x_i^{(1)}} \cdot n_i^{(1)} - \frac{\partial \hat{x}_i^{(2)}}{\partial x_i^{(2)}} \cdot n_i^{(2)}.$$

Compared with Case (1), there now exists an intrinsic target inconsistency term $(s_i^{(1)} - s_i^{(2)})$ that does not vanish even if the denoiser perfectly suppresses the noise-sensitive terms, making the problem fundamentally ill-posed due to the lack of a unique underlying target. Therefore, minimizing the pairwise consistency loss no longer uniquely drives the outputs toward a shared clean surface point, but instead forces the network to compromise between inconsistent targets. This introduces an irreducible bias in the optimization objective and leads to biased and unstable convergence.

**Case (3): Non-symmetric perturbation under deterministic target.**

Here, we consider the non-symmetric case as $\tilde{n}_i = -n_i + \delta_i$, where $\delta_i$ denotes the deviation from ideal mirror symmetry.

Substituting $\tilde{n}_i = -n_i + \delta_i$ into Taylor expansion around the same target surface point $s_i$, we obtain

$$\hat{x}_i - \bar{x}_i = \frac{\partial \hat{x}_i}{\partial x_i} \cdot n_i - \frac{\partial \bar{x}_i}{\partial \tilde{x}_i} \cdot (-n_i + \delta_i) + \mathcal{O}(n_i^2, \tilde{n}_i^2),$$

which can be simplified as

$$\hat{x}_i - \bar{x}_i \approx \left( \frac{\partial \hat{x}_i}{\partial x_i} + \frac{\partial \bar{x}_i}{\partial \tilde{x}_i} \right) \cdot n_i - \frac{\partial \bar{x}_i}{\partial \tilde{x}_i} \cdot \delta_i.$$

Compared with Case (1), where ideal symmetry removes target ambiguity and isolates a pure first-order noise-sensitive term, which can then be suppressed by MPCL, the additional residual term

$$- \frac{\partial \bar{x}_i}{\partial \tilde{x}_i} \cdot \delta_i$$

cannot be eliminated, as it directly depends on the asymmetry error $\delta_i$ from mirror symmetry, introducing an irreducible perturbation in the objective. This theoretically shows that non-optimal or learnable $w_2$ is unfavorable for convergence, as it may be break symmetry and be affected by local uncertainty, degrading convergence compared to the deterministic symmetric setting in Case (1).

