# OpenReview forum: "SIMPC: Learning Self-Induced Mirror-Point Consistency for Unsupervised Point Cloud Denoising"
_ICML.cc/2026/Conference — ICML 2026 regular_

### Official Review · Reviewer_fux9 · 2026-03-04

**Soundness:** 3
**Presentation:** 3
**Significance:** 3
**Originality:** 3
**Overall Recommendation:** 5
**Confidence:** 4

**Summary:**

This paper introduces a novel unsupervised framework for denoising point clouds, named SIMPC. For a given 3D shape _S_, SIMPC (i) generates _M_ noisy variants, (ii) maps each into a latent space via independent encoders, and finally (iii) applies a shared denoiser for iterative refinement. To ensure the denoised points accurately align with the original underlying surface of _S_, the authors introduce a custom point-level loss component (MPCL), which actively minimizes the difference between denoised points and their mirrored counterparts, generated via a novel Mirror-Point Generation module. Extensive results within 3D data highlight the quality and robustness of the proposed approach.

**Compliance With Llm Reviewing Policy:**

Affirmed.

**Final Justification:**

As stated in my review, I find the presented approach SIMPC truly interesting. My questions have been adequately addressed in the rebuttal, and the authors have promised to include more robust theoretical convergence guarantees as well as an analysis of convergence speed. My only concern, as previously stated, lies in the lack of comparison with UCAN, given that it is a very recent method with strong performance on the problem of unsupervised point cloud denoising. In this regard, I strongly recommend making the code used for SIMPC open source, in order to avoid such issues in the future.

**Key Questions For Authors:**

1. In section 4, the authors explicitly cite **U-CAN** [1] as the basis for their unsupervised denoising approach, despite subsequently employing different denoising strategies. Surprisingly, however, the experimental section does not present any **qualitative** comparison between SIMPC and U-CAN, not even on the standard PUNet dataset. U-CAN results are only reported in Table 1, where it generally represents the second best method after SIMPC. To strengthen the evaluation, the authors should include these visual comparisons at least in **Figures 4** and **7**.

2. Throughout the paper, authors frequently refer to the _underlying surface_ of the shape $\mathcal{S}$. However, the paper proposes an unsupervised approach to the denoising problem. Given that the proposed SIMPC framework is unsupervised, a natural question arises to me: how does MPGM guarantee convergence to the _true target surface_ of $\mathcal{S}$? Consequently, the geometric intuition behind setting $w_2=2$ is too rushed. Specifically, why this exact value would project the denoised point $\hat{x}_i$ to the **opposite side of the underlying surface**? Any value of $w_2$ would project the point to the "other side" of the surface..

3. Is it possible to provide more insight into the choice of $L=2$ denoiser blocks?

 \
[1] Zhou, J., Shi, X., Song, H., Fang, Y., Liu, Y. S., & Han, Z. (2025). U-CAN: Unsupervised point cloud denoising with consistency-aware noise2noise matching. arXiv preprint arXiv:2510.25210.

**Limitations:**

Yes

**Strengths And Weaknesses:**

The proposed method (SIMPC) for shape (as point clouds) denoising is highly promising, well motivated and is definitely a contribution in the field of point clouds denoising. The related works are recent and thoroughly discussed both in the methodology and the experimental sections. The key contribution lies in the introduction of a Mirror-Point Generation module (MPGM), which generates "mirrored" versions of denoised points. The idea of encouraging consistency between denoised and mirrored denoised points by minimizing the MPC loss is intuitive and fascinating; it proves to be a fundamental key for driving the convergence toward the shape's underlying surface. Nevertheless, I will later express some uncertainties and doubts that have arisen. Overall the paper is well written, though there are some notational inconsistencies that cause confusion.

**Theoretical guarantees of the MPGM & MPC Loss:** As noted by the authors in Section 4, SIMPC utilizes an architectural structure similar to several previous state-of-the-art approaches. The key innovation is introducing MPG module coupled with MPC loss to force noisy points toward an _expected_ underlying surface. Given that this strategy is completely new (intuitive and well motivated) and yields very encouraging improvements in quantitative/qualitative results, the paper currently lacks a rigorous analysis regarding theoretical guarantees and convergence speeds, e.g. how close and how fast (depending on hyper-parameters) the _expected surface_ would converge to the true _target surface_ of $\mathcal{S}$.

**Geometric interpretation of MPGM:** Following the previous point, it is crucial to investigate the geometric behaviour induced by the hyper-parameter $w_2$​ in the MPG module. Indeed, the positioning of the mirrored points (which depends on $w_2$) defines an implicit target for denoising. In this sense, the ablation study conducted on $w_2$​ feels somewhat rushed and lacks theoretical justification.

**Experimental methodology:** The authors present multiple experiments where SIMPC consistently emerges as the most robust solution. However, I have few concerns regarding the evaluation protocol:
- the results reported in the tables appear to be derived from single runs, as there are no means or stds provided;
- authors do not explicitly report whether the results for the state-of-the-art methods come from new runs or if they were directly copied from the original papers (I suspect the latter);
- as explicitly demanded in Question 1 in later, there are no qualitative comparisons between SIMPC and U-CAN, which is a highly relevant baseline for unsupervised point cloud denoising.

## Some minor remarks
**Notational Inconsistencies:** In Section 3, the notation used to define the independent sampled noisy clouds ($X^m$) conflicts with subsequent notation. Specifically, these samples are later denoted using subscript notation ($X_a​$,$X_b$​, etc.), whereas superscript is then used to denote the progressive denoising steps ($X^0,\dots,X^L$). I recommend standardizing the notation to ensure better clarity and consistency throughout the manuscript. \
(Furthermore, the paper lacks a running title)

---

> ### Author Rebuttal · Authors · 2026-03-31
>
> We deeply appreciate the reviewer fux9 for the thoughtful feedback and
> time invested in evaluating our work.
> We respond to each question below.
>
> ## **Q1: Theoretical Analysis of how close and how fast of SIMPC**
>
> (1) How Close.
>
> Theoretical analysis showing how MPCL promotes consistent denoising targets is in Appendix.A.4.
> To further address the reviewer’s concern, we extend it to GT surface by following the analytical
> framework of SIMPC.
>
> ### Case (1): Symmetry and deterministic correspondence for GT surface
>
> For a noisy seed point $x_i$, let $s_i \in \mathcal{S}$ denote its corresponding target point
> on the underlying clean surface $\mathcal{S}$, and let the noise be written as $x_i = s_i + n_i.$
> At the current denoising stage, let the denoiser output a point-wise denoising vector $d_i = \mathcal{D}(x_i),$
> so that the normally denoised seed point is $\hat{x}_i = x_i + d_i.$
> We further construct a mirror-point by extending the denoising vector $\tilde{x}_i = x_i + 2 d_i,$
> and denote its denoised mirror-point by
> $\bar{x}_i = \tilde{x}_i + \tilde{d}_i$.
>
> The Mirror-Point Consistency Loss is then
> $$
> \mathcal{L}_{\mathrm{MPC}} = \mathbb{E}\left[\|\hat{x}_i - \bar{x}_i\|^2\right].
> $$
>
> Through Taylor expansion, we can approximate the denoised seed point and
> the denoised mirror-point around the target surface point $s_i$:
> $$
> \hat{x}_i
> = s_i + \frac{\partial \hat{x}_i}{\partial x_i} \cdot n_i + \mathcal{O}(n_i^2),
> $$
> $$
> \bar{x}_i
> = s_i - \frac{\partial \hat{x}_i}{\partial x_i} \cdot n_i + \mathcal{O}(n_i^2).
> $$
>
> Substituting the approximations, we get
> $$
> \hat{x}_i - \bar{x}_i
> \approx
> 2*\frac{\partial \hat{x}_i}{\partial x_i} \cdot n_i.
> $$
>
> With further Taylor expansion, the loss function can be simplified as
> $$
> \mathcal{L}_{\mathrm{MPC}}
> \approx
> 4*\left\|
> \frac{\partial \hat{x}_i}{\partial x_i}\cdot n_i
> \right\|^2.
> $$
>
> Through the consistency loss $\mathcal{L}_{\mathrm{MPC}}$, we optimize the network such that
> $$
> \frac{\partial \hat{x}_i}{\partial x_i}\cdot n_i \rightarrow 0,
> $$
> which implies that the denoising process becomes locally insensitive to symmetric perturbations around the same target surface point.
>
> Through the consistency loss $\mathcal{L}_{\mathrm{MPC}}$, we optimize the network
> $\mathcal{D}$ such that $\frac{\partial \hat{x}_i}{\partial x_i} \cdot n_i$
> is driven towards zero, which means that the output $\hat{x}_i$ tends to:
> $$
> \hat{x}_i = \mathcal{D}(x_i)
> = \mathcal{D}(s_i + n_i)
> $$
> $$
> = \mathcal{D}(s_i)+
> \frac{\partial \hat{x}_i}{\partial x_i} \cdot n_i+
> \mathcal{O}(n_i^2)
> $$
> $$
> \approx \mathcal{D}(s_i).
> $$
>
> Under the ideal assumption that the denoiser has learned the geometric prior
> of the underlying surface, we further have $\mathcal{D}(s_i) \approx s_i.$
>
> Thus, the final denoised output satisfies $\hat{x}_i \approx s_i,$
> and similarly $\bar{x}_i \approx s_i.$
>
> Therefore, the denoised seed point and its mirror counterpart converge to the same target surface point:
> $$
> \hat{x}_i \approx \bar{x}_i \approx s_i \in \mathcal{S}.
> $$
>
> In summary, the favorable convergence of SIMPC relies on two key conditions:
> deterministic correspondence and symmetry with respect to the same target surface.
> MPCL primarily suppresses the noise-sensitive component and
> drives both outputs toward a shared clean surface point.
> This behavior is consistent with the maximum-likelihood assumption
> in score-based denoising, where the ground-truth location is inferred unsupervised
> from noisy observations; under zero-mean Gaussian noise,
> this typically corresponds to the underlying clean surface.
>
> (2) How Fast.
>
> ***Tab C. Convergence speed under different numbers of denoiser blocks $L$.***
>
> |Method|10K,1%||10K,2%||10K,3%||
> |-|-|-|-|-|-|-|
> |Metric|P2M|CD|P2M|CD|P2M|CD|
> |SIMPC($L=1$)|23.99|4.80|31.57|8.46|41.64|16.30|
> |SIMPC($L=2$)|20.25|3.60|28.82|7.13|34.42|13.85|
> |SIMPC($L=4$)|20.01|3.62|28.29|6.89|33.95|13.61|
>
> We analyze convergence speed with respect to hyperparameters
> (e.g. numbers of denoiser blocks), as in Tab. C,
> Increasing $L$ yields only marginal gains, while **$L=2$ already achieves stable and fast convergence.**
>
> ---
>
> ## **Q2: Geometric interpretation of MPGM**
>
> A fixed $w_2 = 2$ enforces symmetric extension of deterministic target,
> ensures stabilizes MPCL optimization as discussed in R1.
> More results on symmetric and cross surface guarantee is in *Reviewer KgFj's Fig. B.*
>
> ---
>
> ## **Q3: Experimental Methodology**
>
> **We strictly follow the evaluation protocols adopted in prior works.**
> Since the performance variance is negligible,
> mean and std is generally not included.
> PUNet is a widely used so we directly cite the results in previous papers.
> We reproduce Noise2Score3D, DMR-U, and Score-U
> on PCNet and other non-Gaussian and real-world datasets.
> As **U-CAN have no code (project page exists, empty code link)**,
> we are unable to compare on other datasets or modalities.
> **We train SIMPC with the same loss as U-CAN, as in Tab. 5 with $L_{SR(EMD)}$,
> SIMPC still demonstrates clear advantages.**

---

> > ### Author Rebuttal · Reviewer_fux9 · 2026-04-02
> >
> > All my concerns are well addressed and I upgrade my note.

---

> > > ### Author Response · Authors · 2026-04-07
> > >
> > > We sincerely thank you for your positive feedback and helpful suggestions.  We will incorporate theoretical guarantees for accurate convergence under deterministic correspondence and mirror symmetry designs, and provide an analysis of convergence speed across iterations.  We will also supplement implementation and evaluation details for our comparisons.

---

### Official Review · Reviewer_WJAr · 2026-03-06

**Soundness:** 3
**Presentation:** 2
**Significance:** 3
**Originality:** 3
**Overall Recommendation:** 5
**Confidence:** 3

**Summary:**

This manuscript proposes to generate mirror-point and construct one-to-one deterministic correspondences to help point cloud denoising. Mirror-point generation module (MPGM) is designed to extract geometric priors, and Mirror-point consistency loss (MPCL) is adopted to encourage consistency between the original denoising points and its mirror counterpart. The key idea is that accurate estimation lead to small MPCL. Promising experimental results are reported.

**Compliance With Llm Reviewing Policy:**

Affirmed.

**Final Justification:**

It is my belief that it is a technically solid paper, with high impact on at least one sub-area of AI, with good-to-excellent evaluation, resources and reproducibility.

**Key Questions For Authors:**

1. Why correspondence ambiguity is bad? What is the role and importance of deterministic one-to-one correspondences? What is the superiority of deterministic correspondences over ambiguous ones?
2. Do symbols T (line 158) and L (line 210）represent the same concept? If so, they can be merged for clarity. If not, please clarify their difference.
3. Does “Decoder” means the "Denoiser Block" in Figure 2? or “Decoder” is part of the Denoiser Block? It should be clearly illustrated in Figure 2.
4. Why deterministic correspondences lead to superior performance? Only experimental support is not sufficient. It would be better to also provide theoretical support. The theory in the appendix is awful for ordinary readers. Some brief and easy-to-understand arguments are preferred.

**Limitations:**

Limitation is missing. I do not believe there are no failure cases.

**Strengths And Weaknesses:**

Pros:
1. A simple and interesting point cloud denoising framework by smartly usage of the mirror principle.
2. Promising results and convincing ablations.

Cons:
1. The figures are weak and difficult to follow.
- Figure 1: a) All kinds of arrows and circles diffuse the readers without necessary explanation.
         b) What is the difference between the circles with and without outline?
- Figure 2: If possible, notations like U, G and F should be illustrated in Figure 2.
- Figure 3: Can we take PSA+Decoder=Denoiser Block?
- Figure 4: What is the difference between red and green dashed box?

2. Missing limitation is a big limitation. Failure cases and applicable scenarios should be reported.

---

> ### Author Rebuttal · Authors · 2026-03-31
>
> We deeply appreciate the reviewer WJAr for the thoughtful feedback and
> time invested in evaluating our work.
> We respond to each question below.
>
> ## **Q1: Detailed Explanation of Fig. 1**
>
> In Fig. 1, circles represent noisy and denoised points (or pixels),
> while arrows indicate the directions of noise perturbation and denoising vectors, respectively.
> Circles with and without outlines are used to distinguish **different noisy variants.**
> We will add the necessary explanations and clarifications in the revised version.
>
> ---
>
> ## **Q2: Feature Notations in Fig. 2**
>
> $G$ denotes the features inside the Encoder.
> $F$ denotes the features inside the Denoiser Block.
> $U$ denotes the features between the Denoiser Blocks.
>
> We will include these notations in the revised Fig. 2.
>
> ---
>
> ## **Q3: Clarification of Feature Notations in Fig. 3**
>
> In the upper part of Fig. 3, the Denoiser Block consisting of PSA + Decoder,
> with input channel dimension C as defined in Eq. (6).
> In the lower part of Fig. 3, the PSA module is not shared with the Denoiser Block,
> and the input channel dimension is C+3, as defined in Eq. (9).
>
> We will clarify this distinction in the revised version.
>
> ---
>
> ## **Q4: Difference Between Red and Green Dashed Boxes in Fig. 4**
>
> In Fig. 4, supervised methods is in red and unsupervised methods in green.
> We will add explicit clearer color annotations in the revised figures.
>
> ---
>
> ## **Q5: Limitations and Failure Cases of SIMPC**
>
> We observe that SIMPC exhibits **slight performance degradation under extremely sparse point cloud and geometry smooth.**
> Details please refer to *Reviewer HTir' R2*.
>
> ---
>
> ## **Q6: Meaning of Symbols $T$ and $L$**
>
> $T=3$ and $L=2$ represents the number of layers in the DGCNN encoder and Denoiser Blocks.
>
> ---
>
> ## **Q7: Why deterministic correspondences lead to superior performance**
>
> Following the definition in *Reviewer fux9's Q1*,
> we present theoretical analysis explaining
> why ambiguous correspondence is not better than deterministic one.
>
> ### Case (2): Ambiguous correspondence with inconsistent denoising targets
>
> The derivation in Case (1) is based on the fact that the two points
> in the consistency pair correspond to the same underlying surface target.
> If the correspondence is ambiguous (constructed by noise-based or EMD-based methods),
> the two noisy points no longer share the same clean target.
> Let the paired points be associated with two different surface points $s_i^{(1)}$ and $s_i^{(2)}$
> and independent noise condition, with
> $$
> x_i^{(1)} = s_i^{(1)} + n_i^{(1)},\qquad x_i^{(2)} = s_i^{(2)} + n_i^{(2)},\qquad s_i^{(1)} \neq s_i^{(2)}.
> $$
> Then their denoised outputs can be approximated as
> $$
> \hat{x}_i^{(1)} = s_i^{(1)} + \frac{\partial \hat{x}_i^{(1)}}{\partial x_i^{(1)}} \cdot n_i^{(1)} + \mathcal{O}((n_i^{(1)})^2),
> $$
> $$
> \hat{x}_i^{(2)} = s_i^{(2)} + \frac{\partial \hat{x}_i^{(2)}}{\partial x_i^{(2)}} \cdot n_i^{(2)} + \mathcal{O}((n_i^{(2)})^2).
> $$
> Their difference becomes
> $$
> \hat{x}_i^{(1)} - \hat{x}_i^{(2)} \approx
> (s_i^{(1)} - s_i^{(2)}) +
> \frac{\partial \hat{x}_i^{(1)}}{\partial x_i^{(1)}} \cdot n_i^{(1)} -
> \frac{\partial \hat{x}_i^{(2)}}{\partial x_i^{(2)}} \cdot n_i^{(2)}.
> $$
> Compared with Case (1), there now exists an intrinsic target inconsistency term $(s_i^{(1)} - s_i^{(2)})$
> that does not vanish even if the denoiser perfectly suppresses the noise-sensitive terms,
> **making the problem fundamentally ill-posed due to the lack of a unique underlying target**.
> Therefore, minimizing the pairwise consistency loss no longer uniquely drives the outputs
> toward a shared clean surface point,
> but instead forces the network to compromise between inconsistent targets.
> This introduces an irreducible bias in the optimization objective and leads to biased and unstable convergence.

---

> > ### Author Rebuttal · Reviewer_WJAr · 2026-04-01
> >
> > My concerns are well addressed.

---

> > > ### Author Response · Authors · 2026-04-07
> > >
> > > We sincerely thank you for your positive feedback and helpful suggestions.  We will improve the readability of the figures by refining the symbols and captions.  We will also incorporate theoretical guarantees for establishing deterministic one-to-one correspondences, and add a discussion of failure cases and generalization boundaries of SIMPC in the limitations section.

---

### Official Review · Reviewer_KgFj · 2026-03-09

**Soundness:** 2
**Presentation:** 3
**Significance:** 3
**Originality:** 2
**Overall Recommendation:** 4
**Confidence:** 3

**Summary:**

To address the correspondence ambiguity problem inherent in unsupervised point cloud denoising tasks, this paper proposes a novel framework based on Self-Induced Mirror-Point Consistency (SIMPC). Distinct from prior approaches that rely on random noise injection (e.g., Noise2Score) or Earth Mover's Distance (EMD) for distribution alignment, this method generates a mirror point on the opposite side of the underlying surface via linear extrapolation ($w_2=2$) along the predicted denoising direction. It then enforces the denoising predictions of both the original point and its mirror counterpart to converge toward the identical surface location. The proposed method demonstrates highly competitive performance on both synthetic datasets and real-world scans, significantly outperforming existing unsupervised baselines, particularly regarding the Point-to-Mesh (P2M) distance metric.

**Compliance With Llm Reviewing Policy:**

Affirmed.

**Final Justification:**

All my concerns are well addressed, and I will raise my score.

**Key Questions For Authors:**

1) Please provide a denoising error evaluation stratified by local curvature levels. Does SIMPC exhibit significantly higher Point-to-Mesh (P2M) errors in high-curvature regions compared to locally flat areas? Given that the linear extrapolation $w_2=2$ is geometrically precise only under a local flatness assumption, it is crucial to quantify the model's performance degradation on sharp edges and corners.

2) How can it be demonstrated that the neighborhood information extracted from an empty perspective (i.e., the void side of a single-sided scan) is truly constructive for denoising, rather than merely acting as a form of data augmentation or stochastic perturbation? The authors are encouraged to provide a visualization (e.g., t-SNE or PCA) of the refined mirror features $\tilde{f}_i$ to show whether they occupy a consistent manifold with the original features or if they primarily serve as a regularization bottleneck.

3) Since a fixed scaling factor of $w_2=2$ is only a first-order approximation for flat surfaces, have the authors explored allowing the network to predict an adaptive $w_2$? Alternatively, have you considered dynamically adjusting the step size based on local geometric properties, such as the local covariance matrix or curvature estimates? Such an analysis would clarify whether the current heuristic is a fundamental limitation or a deliberate trade-off for computational efficiency.

**Limitations:**

Lack of Discussion on Limitations and Negative Impacts

**Strengths And Weaknesses:**

Strengths:
1) This paper departs from the conventional global/local point-set distribution matching paradigm prevalent in unsupervised denoising. By leveraging the network's own predicted denoising vector $d_i$ to construct deterministic one-to-one correspondences, it introduces an effective and intuitive heuristic.

2) It is computationally efficient, effectively circumventing the substantial computational overhead and overfitting risks associated with optimal transport algorithms such as EMD.

Weaknesses:
1) algorithm relies heavily on a scalar multiplication of $w_2=2$ within Euclidean space. While this is equivalent to a normal mirror reflection in locally flat regions, it presents significant issues in high-curvature areas—such as the sharp edges and corners of CAD models. In these cases, the linear extrapolation $\tilde{x}_i = x_i + 2d_i$ causes the generated mirror point $\tilde{x}_i$ to deviate severely from the surface's normal symmetry axis, essentially shooting it into invalid free space. Although the paper demonstrates the global optimality of $w_2=2$ through ablation studies, it fails to investigate the sub-optimal performance of this approach in regions characterized by abrupt local geometric transitions.

2) Logical Discontinuity in Feature Space: The authors directly assign the high-dimensional feature $u_i$ of the original point to the mirror point $x_i$ at a completely different spatial location, subsequently performing feature fusion by extracting a new neighborhood via KNN. In real-world single-sided scans, the opposite side of the surface is essentially a void (an empty perspective). The KNN neighborhood forcibly sought across the surface boundary exhibits a massive domain shift in its topological distribution compared to that of the original point. Furthermore, the direct reuse of $u_i$ lacks a rigorous proof of equivariance. Consequently, from a mathematical standpoint, this module appears more as a highly robust noise regularizer rather than a true mirror feature refinement mechanism.

3) Unstable Gradients at Cold Start: During the initial epochs of model training, the predicted denoising vector $d_i$ approximates a random distribution. Consequently, the generated mirror points $x_i$ are scattered throughout the 3D space, causing the KNN search to capture entirely irrelevant local topologies. This, in turn, propagates violent and erroneous gradients back through the network via $L_{MPC}$. The paper provides no explanation of how this catastrophic error during the early stages of non-convex optimization is mitigated—for instance, through the use of a warm-up phase or a dynamic $w_2$ scheduling strategy.

4) Lack of Discussion on Limitations and Negative Impacts

---

> ### Author Rebuttal · Authors · 2026-03-31
>
> We deeply appreciate the reviewer KgFj for the thoughtful feedback and
> time invested in evaluating our work.
> We respond to each question below.
>
> ## **Q1: P2M in sharp and flat**
>
> ***Fig A. Examples for region split.***
> (https://anonymous.4open.science/r/Anonymize_SIMPC-F112/fig1.png)
>
> We use PCA to split CAD models in PUNet into most sharp (10%)
> and flat (90%) regions, as shown in Fig. A.
>
> ***Tab C. P2M in Sharp and Flat regions.***
>
> ||Dataset:PU|10K,1%||10K,2%||10K,3%||
> |-|-|-|-|-|-|-|-|
> ||Methods|Sharp|Flat|Sharp|Flat|Sharp|Flat|
> |E0|Noisy|10.78|12.32|35.98|40.63|79.40|85.38|
> |E1|StraightPCF|4.05|3.29|6.71|5.05|10.38|8.18|
> |E2|SIMPC($L_{MPC}+L_{SR(CD)}$)|4.57|3.69|7.89|6.10|11.73|8.52|
> |E3|SIMPC($\mathcal{L}_{\mathrm{SR(EMD)}}$)|6.54|5.77|14.33|12.64|19.33|18.89|
> |E4|Noise2Score3D|8.74|7.98|18.26|16.42|24.65|24.88|
> |E5|SIMPC($MirrorPoint$)|11.32|13.14|37.80|42.13|83.27|88.99|
>
> As shown in Tab. C (E1–E4) and Fig. C, **SIMPC (E2) achieves consistent and significant improvements
> in both sharp and flat regions** over EMD-based (E3) and noise-based (E4) methods,
> and performs comparably to the SOTA supervised method (E1).
>
> ---
>
> ## **Q2: Distribution of Mirror Points**
>
> ***Fig B. Visualization on noisy and mirror points with extend denoising vector and cross surface ratio.***(https://anonymous.4open.science/r/Anonymize_SIMPC-F112/fig2.png)
>
> **Mirror points lie near the surface with stable geometry symmetry and
> similar distribution with seed points.**
> The P2M of mirror and seed points is close, as in Tab. C (E0, E5).
> Most seed points (90%+) arrived to the other side and
> denoising vector well-aligned with surface normal, as in Fig. B.
> **This establishes clear geometric constraints rather than using perturbation enhancement to facilitate denoising convergence.**
>
> ---
>
> ## **Q3: Single-Sided Scan**
>
> The ranging noise of visible surfaces can be
> viewed as a Gaussian noise around a thin manifold, and
> **does not exhibit a dense side versus a void side**.
> Invisible surfaces need completion model,
> which is out the scope of this work.
> **Our evaluation on real-world single-sided scans in Figs. 5, 6, and 8
> and Tab. 5 demonstrates stable results.**
>
> ---
>
> ## **Q4: Representation Consistency of Seed and Mirror Point**
>
> ***Fig C. t-SNE of Seed and Mirror Point Features.***
> (https://anonymous.4open.science/r/Anonymize_SIMPC-F112/fig3.png)
>
> We clarify that $u_i$ serves for initialization rather than equivariance.
> The refined feature $\tilde{f}_i$ and $u_i$ correspond to observations of the same surface at different positions.
> Fig. C shows that $\tilde{f}_i$ and $u_i$ are close in t-SNE space for the same point,
> indicating **consistent manifold representations without domain shift.**
>
> ---
>
> ## **Q5: Applicability of Symmetry in Sharp Regions**
>
> ***Fig D. Experiment on Corner Denoising.***
> (https://anonymous.4open.science/r/Anonymize_SIMPC-F112/fig4.png)
>
> Denoising methods, even supervised, tend to smooth corners into
> curved continuous surfaces and can be locally approximated to a plane,
> which makes the assumption **still valid for sharp regions**, as in Fig. B and Fig. D.
>
> ---
>
> ## **Q6: Fixed $w_2=2$ vs. Other Values and Learnable Strategy**
>
> Following the definition in *Reviewer fux9's Q1*,
> we present theoretical analysis on why $w_2=2$ is best.
>
> ### Case (3): Non-symmetric perturbation under deterministic target.
> Here, we consider the non-symmetric case as $\tilde{n}_i = -n_i +\delta_i,$
> where $\delta_i$ denotes the deviation from ideal mirror symmetry.
>
> Substituting $\tilde{n}_i = -n_i + \delta_i$ into Taylor expansion around
> the same target surface point $s_i$, we obtain
> $$
> \hat{x}_i - \bar{x}_i =
> \frac{\partial \hat{x}_i}{\partial x_i}\cdot n_i -
> \frac{\partial \bar{x}_i}{\partial \tilde{x}_i}\cdot(-n_i + \delta_i) +
> \mathcal{O}(n_i^2, \tilde{n}_i^2),
> $$
> which can be simplified as
> $$
> \hat{x}_i - \bar{x}_i
> \approx
> \left(
> \frac{\partial \hat{x}_i}{\partial x_i} +
> \frac{\partial \bar{x}_i}{\partial \tilde{x}_i}
> \right)\cdot n_i -
> \frac{\partial \bar{x}_i}{\partial \tilde{x}_i}\cdot \delta_i.
> $$
>
> Compared with Case (1), where ideal symmetry removes target ambiguity and isolates a
> pure first-order noise-sensitive term, which can then be suppressed by MPCL,
> the additional residual term
> $$
> -\frac{\partial \bar{x}_i}{\partial \tilde{x}_i}\cdot \delta_i
> $$
> cannot be eliminated, as it directly depends on the asymmetry error $\delta_i$ from mirror symmetry,
> introducing an irreducible perturbation in the objective.
> **This theoretically shows that non-optimal or learnable $w_2$ is unfavorable for convergence,
> as it may be break symmetry and be affected by local uncertainty,
> degrading convergence compared to the deterministic symmetric setting in Case (1).**
>
> ---
>
> ## **Q7: Training Stability at Early Stage**
>
> **We didn't observe convergence issues even without any warm-up tricks.**
> The loss stabilizes within ~100 iters and converges in ~30 epochs,
> while using $EMD$ and $CD$ requires ~100 epochs.

---

> > ### Author Rebuttal · Reviewer_KgFj · 2026-04-03
> >
> > Thanks for the author's response. All my concerns are well addressed, and I will raise my score.

---

> > > ### Author Response · Authors · 2026-04-07
> > >
> > > We sincerely thank you for your positive feedback and helpful suggestions. We will incorporate both qualitative and quantitative results on denoising performance in sharp/flat regions, as well as theoretical guarantees of the mirror symmetry design, in the revised manuscript.

---

### Official Review · Reviewer_HTir · 2026-03-13

**Soundness:** 2
**Presentation:** 3
**Significance:** 3
**Originality:** 3
**Overall Recommendation:** 4
**Confidence:** 3

**Summary:**

The paper presents an unsupervised framework for point cloud denoising called SIMPC. The core idea is to use Mirror Points to localize the position of the underlying surface. Specifically, for each noisy point, it self-induces a mirror counterpart on the opposite side of the underlying surface based on a predicted denoising vector. By enforcing a Mirror-Point Consistency Loss, the model encourages both the original point and its mirror point to converge to the same surface location, establishing a deterministic self-consistency mechanism.

**Compliance With Llm Reviewing Policy:**

Affirmed.

**Final Justification:**

My concerns have been adequately addressed. I maintain my positive evaluation.

**Key Questions For Authors:**

Could the authors clarify the generalization boundaries of SIMPC? Specifically, what types of geometries or noise levels is this method adept at handling, and where does it struggle (e.g., in preserving sharp features or handling extremely sparse regions)?

**Limitations:**

Yes

**Strengths And Weaknesses:**

Strengths:

The idea of applying self-induced mirror-point consistency for point denoising is novel. According to the reported experiments, it also achieves better denoising performance than competing methods.

Weaknesses:

However, according to the runtime comparison in Table 6, its inference overhead appears higher than some other methods, such as Noise2Score3D. The analysis of computational complexity is insufficient, as the authors merely present the numerical results but lack a thorough discussion regarding the sources and analysis of this complexity.

---

> ### Author Rebuttal · Authors · 2026-03-31
>
> We deeply appreciate the reviewer HTir for the thoughtful feedback and
> time invested in evaluating our work.
> We respond to each question below.
>
> ## **Q1: Computational Complexity**
>
> ***Tab A. Inference time breakdown.***
>
> |Method| Patch Splitting + Stitching (ms) |Forward(ms)|
> |-|-|-|
> |SIMPC(Ours)|868|143|
> |StraightPCF|1299|161|
> |Noise2Score3D|-|140|
>
> As shown above, the dominant computational complexity of SIMPC
> (and other patch-based methods, e.g., StraightPCF) comes from **patch splitting
> and stitching**, which is a standard processing step for handling
> large-scale point clouds and is not specific to our method.
> This part of the runtime can be **controlled by adjusting the patch size and the number of patches.**
> In contrast, Noise2Score3D does not adopt patch splitting and stitching,
> but directly processes the entire object,
> and therefore its runtime only consists of the forward inference time.
>
> ---
>
> ## **Q2: Generalization Boundaries**
>
> ***Tab B. Comparison on extremely sparse point clouds.***
>
> |Dataset:PU|5K,1%||5K,2%||5K,3%||
> |-|-|-|-|-|-|-|
> |Method|P2M|CD|P2M|CD|P2M|CD|
> |Noise2Score3D|50.31|17.15|69.36|26.83|88.03|45.32|
> |SIMPC|42.85|11.89|58.04|15.77|68.06|21.18|
> |StraightPCF|33.48|6.37|53.39|12.47|64.19|19.32|
>
> The generalization boundaries of SIMPC mainly lie in a **slight performance degradation
> when handling extremely sparse point clouds and a mild smoothing effect on sharp geometric structures**.
>
> We evaluate SIMPC and other methods on extremely sparse point clouds, as shown in Tab. B.
> Compared with the supervised method (StraightPCF), the performance gap of SIMPC slightly increases
> under this setting. Nevertheless, SIMPC still achieves clear improvements over previous
> unsupervised methods such as Noise2Score3D.
>
> In addition, SIMPC achieves performance comparable to supervised methods in flat regions.
> However, for sharp geometric structures, it exhibits slight smoothing compared to supervised approaches.
> This is mainly due to the absence of ground-truth geometric details, which are typically leveraged in
> supervised denoising frameworks. The corresponding visualizations are provided in *Reviewer KgFj’s Fig. D*.
>
> **We will include the analysis of these generalization boundaries in the limitations section.**

---

> > ### Author Rebuttal · Reviewer_HTir · 2026-04-04
> >
> > My concerns have been adequately addressed. I recommend acceptance and will maintain my positive score.

---

> > > ### Author Response · Authors · 2026-04-07
> > >
> > > We sincerely thank you for your positive feedback and helpful suggestions. We will incorporate a discussion on generalization boundaries and an analysis of computational complexity in the revised manuscript.

---

### Decision · Program_Chairs · 2026-04-30

**Decision:**

Accept (regular)

**Comment:**

Eventually, this submission got four positive recommendations. Initially, the reviewers were concerned about the soundness of technical solutions, the presentation, and the runtime complexity. The authors did a good job and addressed most of these concerns in the rebuttal. During the discussion among the authors and the reviewers, the reviewers confirmed that their concerns had been fully addressed. Thus, all reviewers reached a consensus without a discussion. The AC read through the manuscript, all reviews, the rebuttal, and the discussions among the authors and the reviewers, the AC agreed with all reviewers, and liked the idea of the paper. Per these, the AC made a decision of acceptance. This decision was approved by the SAC as well.